# Contribution of GATA6 to homeostasis of the human upper pilosebaceous unit and acne pathogenesis

Bénédicte Oulès[1,2,3,10], Christina Philippeos [1,10], Joe Segal[1,4], Matthieu Tihy [1,5], Matteo Vietri Rudan [1], Ana-Maria Cujba [1], Philippe A. Grange[2,3], Sven Quist[6], Ken Natsuga [7], Lydia Deschamps[8], Nicolas Dupin[2,3], Giacomo Donati [9] & Fiona M. Watt [1✉]

Although acne is the most common human inflammatory skin disease, its pathogenic mechanisms remain incompletely understood. Here we show that GATA6, which is expressed in the upper pilosebaceous unit of normal human skin, is down-regulated in acne. GATA6 controls keratinocyte proliferation and differentiation to prevent hyperkeratinisation of the infundibulum, which is the primary pathological event in acne. When overexpressed in immortalised human sebocytes, GATA6 triggers a junctional zone and sebaceous differentiation program whilst limiting lipid production and cell proliferation. It modulates the immunological repertoire of sebocytes, notably by upregulating PD-L1 and IL10. GATA6 expression contributes to the therapeutic effect of retinoic acid, the main treatment for acne. In a human sebaceous organoid model GATA6-mediated down-regulation of the infundibular differentiation program is mediated by induction of TGFβ signalling. We conclude that GATA6 is involved in regulation of the upper pilosebaceous unit and may be an actionable target in the treatment of acne.

[1] Centre for Stem Cells and Regenerative Medicine, King's College London, Floor 28, Tower Wing, Guy's Hospital, Great Maze Pond, London SE1 9RT, UK. [2] Université de Paris, Institut Cochin, Cutaneous Biology Lab, INSERM U1016, UMR8104, 24 rue du Faubourg St Jacques, 75014 Paris, France. [3] Department of Dermatology, Hôpital Cochin, AP-HP, Groupe Hospitalier Paris Centre Cochin-Hôtel Dieu-Broca, 89 rue d'Assas, 75006 Paris, France. [4] Division of Gastroenterology, University of California San Francisco Liver Center, 513 Parnassus Ave, HSE 1401, San Francisco, CA 94143-1346, USA. [5] Division of Clinical Pathology, University Hospitals of Geneva, Rue Michel-Servet 1, 1206 Genève, Switzerland. [6] Clinic for Dermatology and Venerology, Otto-von-Guericke-University, Leipziger Straße 44, 39120 Magdeburg, Germany. [7] Department of Dermatology, Hokkaido University Graduate School of Medicine N15W7, Sapporo 060-8638, Japan. [8] Department of Pathology, Hôpital Bichat, AP-HP, Hôpitaux Universitaires Paris Nord Val de Seine, 46 rue Henri Huchard, 75018 Paris, France. [9] Department of Life Sciences and System Biology & Molecular Biotechnology Center, University of Turin, Turin, Italy. [10] These authors contributed equally: Bénédicte Oulès, Christina Philippeos. ✉email: fiona.watt@kcl.ac.uk

Acne vulgaris is one of the most common dermatological conditions[1], affecting about 650 million people worldwide, including over 85% of all teenagers. It is a chronic inflammatory disease of the upper pilosebaceous unit, the name given to the hair follicle (HF) and its associated sebaceous gland (SG). Microcomedones and comedones are primary acne lesions that result from occlusion of the HF, preventing sebum from being released onto the skin surface[2]. Hyperseborrhoea and inflammatory papules, pustules, and nodules are also manifestations of acne[3]. Acne places a significant clinical, psychological, and economic burden on patients[4].

The upper pilosebaceous unit includes the infundibulum (INF), which comprises multiple layers of differentiating keratinocytes adjacent to the hair shaft and the interfollicular epidermis (IFE). It also includes the sebaceous gland (SG) and the junctional zone (JZ) at the intersection between the INF, the sebaceous duct (SD), and the isthmus (IST) that lies above the lower, cycling portion of the HF[5] (Fig. 1a). The upper pilosebaceous unit undergoes

discrete changes during the HF cycle[6,7]. Together the INF and JZ provide a channel for the hair shaft as it emerges onto the skin surface[6]. In humans, the SG is a multilobular holocrine gland with an outer layer of proliferative cells surrounding differentiating lipid-rich sebocytes. In mice, SG organisation is similar, although SG are smaller and unilobular, and SD are shorter[8,9]. Within the gland, differentiating sebocytes burst, releasing sebum into the JZ/INF through a short SD to lubricate the skin surface (Fig. 1a). The composition of sebum is species-specific and in humans is mainly composed of triglycerides, free fatty acid, wax esters, squalene, cholesterol esters, and cholesterol[8,9].

SG functions extend beyond sebum production and include the regulation of local androgen and steroid production and synthesis of antioxidants. SG also participate in the immune regulation of pilosebaceous units by secreting proinflammatory and anti-inflammatory molecules[8,10]. A range of different factors control SG homoeostasis, in particular sex steroids and other hormones, neuropeptides, retinoids, PPARγ, and signalling pathways, including

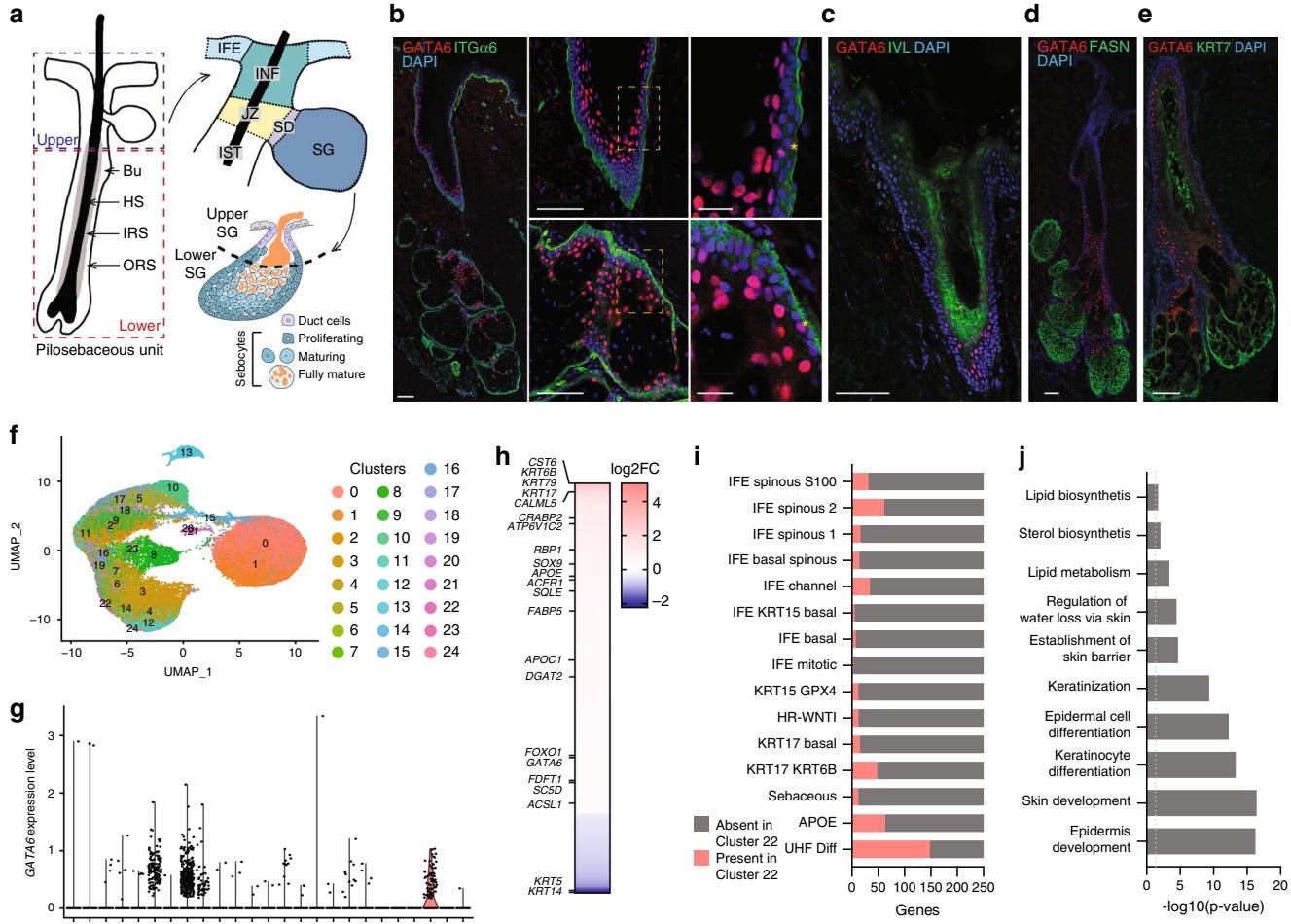

**Fig. 1 GATA6 is expressed in the human upper pilosebaceous unit. a** Schematic of the pilosebaceous unit indicating the upper region, consisting of interfollicular epidermis (IFE), infundibulum (INF), junctional zone (JZ), sebaceous duct (SD), and sebaceous gland (SG), then the isthmus (IST) and the lower region of the hair follicle (HF) comprising the bulge (Bu), hair shaft (HS), inner root sheath (IRS), and outer root sheath (ORS). The lower right panel shows the structure and differentiation of the human SG. **b–e** Immunolabelling for GATA6 (**b–e**) and ITGα6 (**b**), IVL (**c**), FASN (**d**) and KRT7 (**e**), with DAPI counterstain of back skin sections from a 60 year old male. Middle and right panels in **b** are higher magnification views of the left panel section. Yellow asterisks show basal GATA6+ cells. Scale bar: 100 μm, except in **b** right panels: 25 μm. Data are representative of two independent experiments. **f** 2D Uniform Manifold Approximation and Projection (UMAP) visualisation of single cells isolated from healthy human scalp epidermis and adnexa[29] coloured by Seurat cluster. **g** GATA6 expression levels, as log₁₀ Transcripts Per Million (TPM), across the 24 different clusters. GATA6 is mainly expressed in cluster 22. **h** Differentially expressed genes (DEG) of cluster 22. Only DEG with log₂FC > 0.2 or < −0.2 are shown. **i** Intersection of the 250 most upregulated DEG of cluster 22 identified in scalp epidermis within cell populations identified by Cheng et al.[29] HR-WNTI stands for High Resolution-WNT Inhibitory cluster and UHF for upper hair follicle cluster. **j** GO enrichment analysis of the 250 most upregulated DEG of cluster 22.

the Wnt, aryl hydrocarbon receptor, FGFR, ErbB family receptor, and IGF1/insulin pathways[9].

Although acne pathogenesis remains incompletely understood, four main factors have historically been identified as playing a key role. One is hyperkeratinisation of the INF, which leads to HF occlusion. A second is excessive production of sebum under androgen control. Thirdly, there is a disturbance of the INF microbiota, leading to an increase in *Propionibacterium acnes* (*P. acnes*) species (also known as *Cutibacterium acnes*). Finally, release of proinflammatory molecules is linked to acne.

Several observations have shed new light on the physiopathology of acne[3,9,11]. In particular, patients with acne often present with an increase of systemic and local (i.e., produced by the SG) androgen production, leading to an increase of sebocyte proliferation[3]. Additional hormonal imbalance, such as higher levels of steroids and IGF1, has also been observed. Androgens and IGF1 enhance lipid production through upregulation of SREBF1, mTor, and PPAR signalling pathways while inhibiting the FoxO1 transcription factor[9,12]. Acne has also been linked to dysregulation of innate immune signalling in response to *P. acnes* and to increased levels of NFκB-regulated IL1β and IL8 cytokines and of T$_H$17/IL17 T cells[3,13,14]. The role of neuropeptides, in particular substance P, and neurohormones, such as growth hormone (GH) and prolactin, in promoting sebocyte proliferation and sebum production has been shown[3,9,11]. Alteration of fatty acid metabolism, genetic risk factors and environmental cues have also been reported[3,9,13,15]. In addition, the comedone switch hypothesis postulates that there is a common progenitor of the epithelial cells of the INF/JZ/SD and of sebaceous cells, and that this progenitor cell is targeted during comedogenesis and acne[2].

Several studies have provided insights into the development and function of the pilosebaceous unit. Early work described the morphogenesis of pilosebaceous units during human embryonic development[10]. During mouse HF development the progeny of SOX9+ cells initiate SG morphogenesis[16]. LGR6+ mouse epidermal stem cells can also generate sebocytes[17,18]. JUNB expression is associated with a repression of sebaceous fate through NOTCH pathway regulation[19]. LRIG1+ keratinocytes maintain the INF and JZ and can contribute to the upper SG[20–22].

The GATA6 transcription factor is upregulated in the differentiating progeny of LRIG1+ mouse keratinocytes and controls the differentiation of the INF/JZ and upper SG including the SD[23,24]. Retinoids, including isotretinoin, the most effective treatment for acne[25], are known to stimulate expansion of the progeny of LRIG1+ stem cells and also lead to the upregulation of GATA6[20,23]. Upon GATA6 knock-out in KRT5+ keratinocytes, a loss of the ductal cells and a dilation of the INF, JZ, and SD are observed, with cystic lesions filled with keratotic material resembling comedones[23,26]. Nevertheless, acne is a disease specific to humans and no animal model recapitulates its pathological features fully. Therefore, acne research has relied heavily on translational histopathological studies[27] and cell-based studies using human primary cells and immortalised sebocyte cell lines[28].

Since GATA6 plays a role in homoeostasis of the upper HF in mouse, we aimed to explore the role of GATA6 in the human upper pilosebaceous unit by using human samples, 2D, and 3D cellular models and ex vivo organ-cultured hair follicle explants. We postulated that GATA6 could be involved in the differentiation of cells of the INF/JZ/SD on the one hand and of SG on the other hand, as in the comedone switch hypothesis, and could therefore be involved in acne pathogenesis.

In this work, we show that GATA6 expression is reduced in human acne skin. GATA6 controls sebocyte differentiation and proliferation. It is associated with a specific immune signature in sebocytes and has a potential anti-inflammatory role. In addition, we show that GATA6 expression limits the differentiation and proliferation of IFE keratinocytes by promoting TGFβ signalling. Overall, our findings demonstrate that GATA6 is involved in human upper HF homoeostasis and may contribute to acne pathogenesis.

## Results

**GATA6 is a marker of the human upper pilosebaceous unit.** We began by examining GATA6 expression by immunofluorescence profiling of skin from human adults (Fig. 1b–e and Supplementary Fig. 1). As reported previously[23,24], in human as in mouse, GATA6 was expressed in the suprabasal cells of the lower INF, JZ, SD, and of the upper SG. Colocalization with ITGα6 showed that a small number of basal cells were GATA6+, as also found in the mouse[23] (Fig. 1b).

GATA6 was co-expressed with INF differentiation markers such as Involucrin (IVL; also expressed in the IFE) (Fig. 1c) and KRT79 (Supplementary Fig. 1a). The JZ marker PLET1 was predominantly expressed in the lower INF and JZ, where it colocalized with GATA6 (Supplementary Fig. 1b). GATA6 was not expressed in mature lipid-filled sebocytes (Supplementary Fig. 1c). However, GATA6 was co-expressed with some markers of differentiating sebocytes: FASN (Fig. 1d), KRT7 (Fig. 1e), PPARγ (Supplementary Fig. 1d), and BLIMP1 (Supplementary Fig. 1e). The androgen receptor (AR) was expressed in a subpopulation of sebocytes in the lower SG; these cells did not co-express GATA6 (Supplementary Fig. 1f), consistent with the finding that GATA6 represses AR gene expression[23].

There was a strong correlation between regions of LRIG1 and GATA6 expression (Supplementary Fig. 1g), whereas SOX9 was distributed throughout the entire SG and only colocalized with GATA6 in upper sebocytes (Supplementary Fig. 1h). Most GATA6+ cells were nonproliferative, as demonstrated by the absence of costaining for Ki67 (Supplementary Fig. 1i). GATA6 was not expressed in the IFE, sweat glands or lower HF (Supplementary Fig. 1j–m).

Cheng et al. have recently published a single-cell RNA sequencing atlas of epithelial cells originating from three healthy human scalp samples, enabling us to study cells from the pilosebaceous unit[29]. We used the Seurat R package to reanalyse the raw data from this study, performed unbiased clustering, and ran uniform manifold approximation and projection (UMAP) nonlinear dimensional reduction. This approach identified 24 distinct clusters (Fig. 1f). Five hundred and eighty six *GATA6+* cells were identified, while 41,355 cells were negative for *GATA6*. Furthermore, *GATA6+* cells were significantly enriched in cluster 22 (Fig. 1f, g). Fifty-one percent of the cells present in this cluster expressed *GATA6* (Fig. 1g and Supplementary Data 1). To further elucidate the identity of cluster 22 cells, we measured differential gene expression. Several genes of the INF/JZ (*CST6*, *KRT79*, and *ATP6V1C2*) and of SG (*RBP1*, *SOX9*, *SQLE*, and *DGAT2*) were significantly upregulated in this cluster (Fig. 1h and Supplementary Data 1). We then intersected the 250 most upregulated differentially expressed genes (DEG) of cluster 22 and of the different clusters identified in scalp epidermis by Cheng et al.[29] This revealed that cluster 22 mainly corresponds to the "UHF Diff" cluster that contains differentiated upper hair follicle cells (Fig. 1i). Gene ontology (GO) of cluster 22 DEG showed an enrichment for terms relating to epidermis development and differentiation and lipid metabolism (Fig. 1j).

Taken together, the results of the immunohistology staining and single cell RNA sequencing show that GATA6 marks a population of cells of the human upper pilosebaceous unit extending from the lower INF to the upper SG.

**GATA6 expression is reduced in acne skin**. We next examined GATA6 expression in skin biopsies from five healthy controls and nine acne patients presenting with different degrees of disease severity. GATA6 expression was reduced in comedone walls of early acne lesions and was completely lost in advanced lesions (Fig. 2a and Supplementary Fig. 2). The reduction in GATA6 expression occurred regardless of the level of inflammation, as shown by H&E staining. This suggested that the decrease in GATA6 was not a consequence of the increased inflammation in severe acne lesions. Comedone walls were often thicker and more differentiated than the normal upper hair follicle walls, as evidenced by KRT5/6+ cells labelling. We also observed an increase of Ki67+ cells in acne biopsies. Interestingly, the IFE in the vicinity of acne lesions was also thicker and more proliferative than in controls (Fig. 2a and Supplementary Fig. 2).

The reduction in GATA6 expression in acne was further supported by reanalysing published microarray datasets comparing acne skin with matched healthy skin from the same patients or from control patients[30,31]. In both datasets, *GATA6* expression was decreased in acne skin, although the difference was only significant in the larger dataset (Fig. 2b).

We conclude that GATA6 expression is decreased or lost in acne lesions, regardless of their severity.

**GATA6 reduces keratinocyte and sebocyte proliferation**. The observed downregulation of GATA6 in acne samples led us to hypothesise that GATA6 could be involved in the physiological maintenance of the upper pilosebaceous unit in humans. To study this, we inducibly overexpressed GATA6 in primary human IFE and outer root sheath (ORS) keratinocytes, and in the immortalised human sebocyte line SebE6E7. SebE6E7 cells, like the SZ95 line, can differentiate into lipid-filled sebocytes or IVL+ keratinocytes and are thus likely to derive from JZ/SD progenitors that contribute both to the follicular and sebaceous compartments[32].

We cloned the human *GATA6* open reading frame into a lentiviral plasmid with a doxycycline (Dox)-inducible promoter. Infection of primary IFE keratinocytes with GATA6-expressing lentiviruses led to a dose-dependent increase in *GATA6* mRNA (Supplementary Fig. 3a), and to a significant GATA6 upregulation at the protein level (Fig. 3a and Supplementary Fig. 3b, c). We confirmed the correct targeting of GATA6 to cell nuclei (Fig. 3a and Supplementary Fig. 3c). As expected, noninfected and mock-infected IFE keratinocytes did not express GATA6 (Fig. 3a and Supplementary Fig. 3).

INF keratinocytes are hyperproliferative in acne lesions[33]. There was a significant increase in *Ki67* expression in acne samples in published microarray datasets (Fig. 3b), and a higher abundance of basal and suprabasal Ki67+ cells in acne lesions (Fig. 2a and Supplementary Fig. 2). Consistent with these observations, the growth rate of GATA6-expressing ORS and IFE keratinocytes was significantly decreased as compared to controls (Fig. 3c and Supplementary Fig. 4a), and there was a reduction in the number and size of keratinocyte colonies (Fig. 3d and Supplementary Fig. 4b). GATA6 induction also reduced SebE6E7 sebocyte proliferation (Fig. 3e) and decreased colony size in clonogenicity assays (Fig. 3f). We ruled out an effect of GATA6 on apoptosis by performing a TUNEL assay on sebocytes (Fig. 3g).

These results establish that GATA6 negatively regulates proliferation of ORS and IFE keratinocytes and sebocytes.

**GATA6 mediates the JZ and SD differentiation programmes**. In IFE keratinocytes, GATA6 expression led to decreased expression of INF/IFE differentiation markers, IVL and loricrin (LOR) (Fig. 4a). To study the effect of GATA6 on infundibular keratinocytes ex vivo, we cultured whole human hair follicle

explants, as pioneered by the Paus lab[34]. We confirmed GATA6 expression in the lower INF, SD and SG (Supplementary Fig. 5a) of the explants. siRNA-mediated GATA6 knockdown reduced *GATA6* expression compared to scrambled controls (Fig. 4b, Supplementary Fig. 5b) and led to an increase in IVL+ cells in the upper infundibulum (Fig. 4b). However, the effect was modest, most probably because of technical limitations of the model, namely the kinetics of gene knockdown and the time of culture required for hyperkeratinisation to occur.

In IFE keratinocytes, GATA6 expression was not sufficient to induce JZ markers such as *KRT79* and *ATP6V1C2* (Supplementary Fig. 6a). However, when GATA6 was overexpressed in SebE6E7 cells (Supplementary Fig. 6b), several markers of INF/JZ/SD differentiation, including *KRT79*, *IVL*, and *ATP6V1C2*, as well as SG differentiation markers, such as *EMA*, *BLIMP1*, and *CRABP2*, were induced (Fig. 4c). GATA6 also upregulated expression of *LRIG1* and *SOX9*, markers for JZ and SG stem cells, respectively (Fig. 4c).

Expression and activation of PPAR, specifically PPARγ, is required for sebocyte maturation, inducing the synthesis of sebaceous lipids[35]. As PPAR preferentially forms heterodimers with the Retinoid X Receptor (RXR), RXR agonists can synergise with PPARγ agonists to promote sebocyte differentiation[36]. SebE6E7 cells were therefore treated with a PPARγ agonist (troglitazone, TRO) in the presence or absence of a RXR agonist (LG100268, LG). These treatments elicited expression of INF/JZ as well as SG differentiation markers (Supplementary Fig. 6c). Terminal sebaceous differentiation of SebE6E7 cells was obtained upon TRO+LG treatment, as demonstrated by the staining of lipid vacuoles with LipidTOX (Supplementary Fig. 6d). Treatment with TRO or TRO+LG did not significantly increase GATA6-induced differentiation (Supplementary Fig. 6e), suggesting that GATA6 is the principal effector of these treatments.

We also found that GATA6 significantly triggered expression of ductal and sebaceous markers at the protein level by immunostaining for PLET1, KRT7 and SOX9 (Fig. 4d and Supplementary Fig. 6f).

GATA6 overexpression in SebE6E7 cells led to an upregulation of several known SG lipid genes, such as squalene epoxidase (*SQLE*), which is involved in squalene metabolism, and several members of the nuclear receptor superfamily (*LXRβ*, *RARα*, *RXRβ*, *PPARα*, and *PPARγ*)[37,38] (Fig. 4e). Nevertheless, LipidTOX staining revealed decreased numbers of lipid vacuoles in GATA6-expressing sebocytes treated with TRO+LG as compared to controls (Fig. 4f).

These observations suggest that GATA6 is a master regulator of JZ/SG homoeostasis in human, as shown in mouse skin[23]. GATA6 negatively regulates differentiation of IFE keratinocytes but is not sufficient to induce differentiation along the SD lineage. Within sebocytes, it promotes the early phases of differentiation but not the terminal phase when lipid vacuoles accumulate.

**GATA6 mediates the effects of retinoic acid on sebocytes**. Retinoic acid induces the expression of GATA6 in a variety of contexts[39,40] and induces GATA6+ compartment expansion in mouse HF[23]. Isotretinoin (13-cis-retinoic acid) undergoes intracellular isomerisation to all-trans retinoic acid (RA) which then binds to RAR[41]. To explore a potential link between retinoids and GATA6, we first treated SebE6E7 sebocytes with increasing concentrations of RA or 5α-dihydrotestosterone (5αDHT) and performed colony formation assays. While 5αDHT produced a non-significant increase in colony size, treatment with 1 or 5 μM RA phenocopied overexpression of GATA6 by reducing colony size (Fig. 5a). Five micromolar of RA significantly increased GATA6 expression and led to reduced lipid accumulation. In

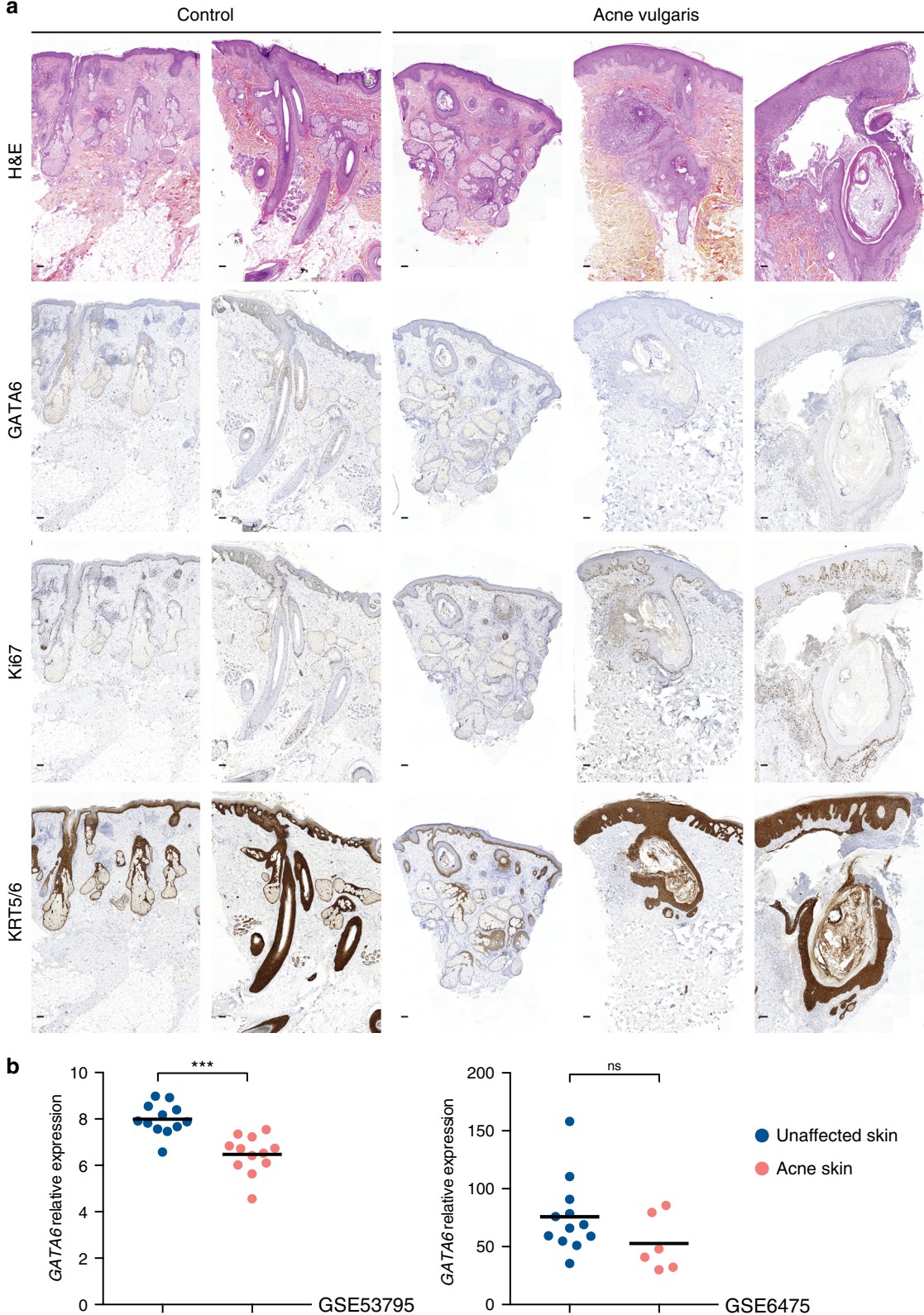

**Fig. 2 Acne skin displays a reduction in GATA6 expression. a** Skin sections from two healthy controls (left panels) and from three acne vulgaris patients presenting with lesions of increasing severity (right panels) stained with haematoxylin and eosin (H&E), or labelled with antibodies against GATA6, Ki67, and KRT5/6 (brown labelling). Scale bar: 100 μm. **b** Expression of *GATA6* transcripts (210002_at probe) was assessed in unaffected skin and acne lesions from microarray datasets GSE53795 ($n = 12$ control skin, $n = 12$ acne skin; $p = 0.00006$) and GSE6475 ($n = 12$ control skin, $n = 6$ acne skin; $p = 0.112$). Mean and individual values are presented. ns not significant; ***p-value < 0.0005; two-tailed unpaired t-test. Source data are provided as a Source Data file.

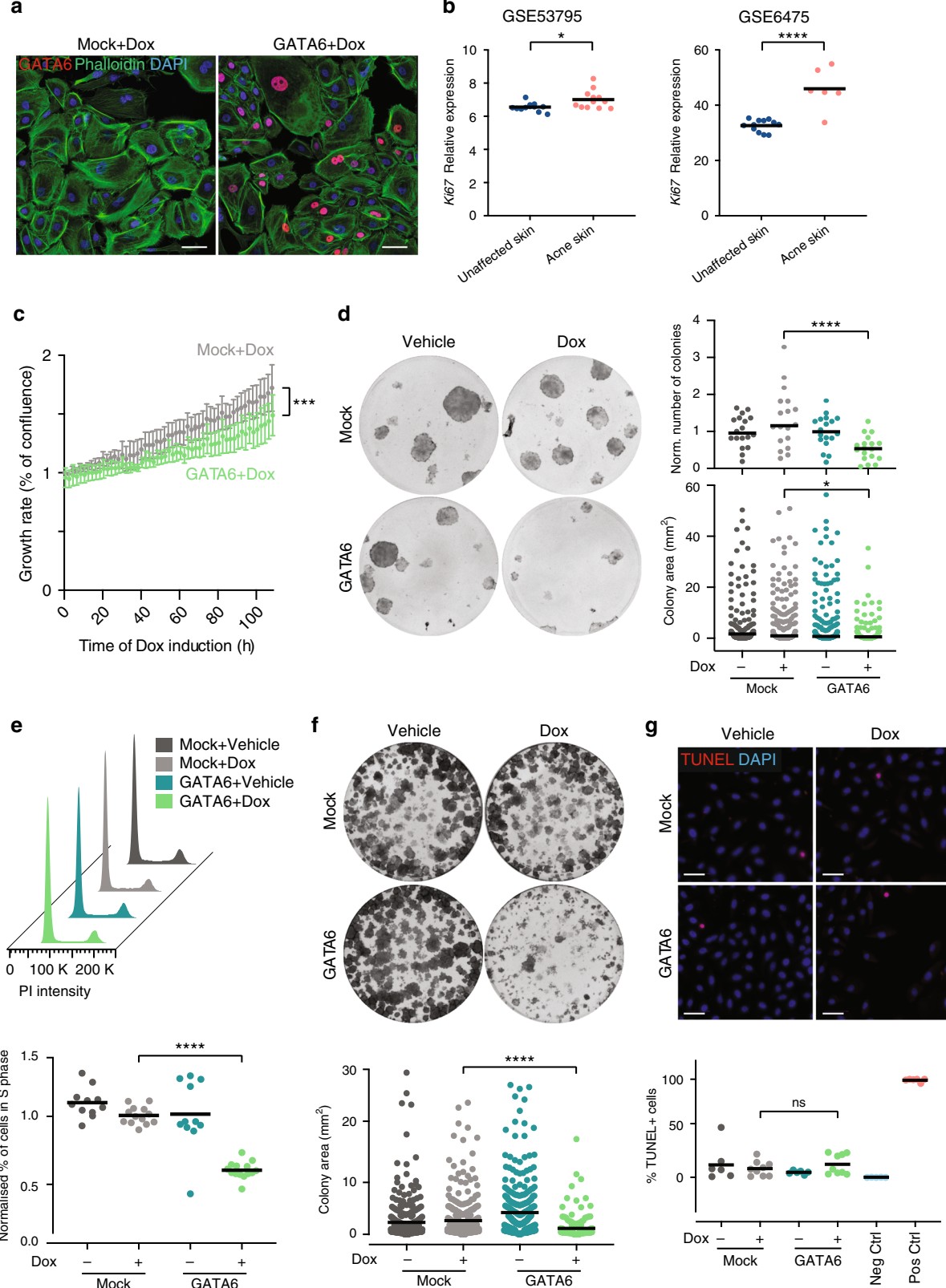

contrast, GATA6 was downregulated in 5αDHT-treated sebocytes and the amount of lipid was significantly increased (Fig. 5b).

To test whether GATA6 could mediate the effects of RA on sebocytes, we used lentiviral vectors to perform Dox-controlled expression of a shRNA targeting human *GATA6* (shGATA6) or a mock sequence (shSCR). Upon addition of Dox, shGATA6

reduced RA-induced *GATA6* mRNA levels (Fig. 5c). In addition, the increase in *EMA* and *PLET1* and decrease in *KRT17* and *cMYC* expression induced by RA were partly reversed upon GATA6 knockdown (Fig. 5d). Lipid production was increased as GATA6 expression was reduced (Fig. 5e) and loss of clonogenicity induced by RA was partially corrected (Fig. 5f).

**Fig. 3 Effects of GATA6 on proliferation, colony formation, and apoptosis. a** Representative immunofluorescence images of Mock and GATA6-infected IFE keratinocytes labelled with antiGATA6 and phalloidin. Cells were treated with 1 μg ml$^{-1}$ doxycycline (Dox) for 16 h. **b** Expression of *Ki67* transcripts (average of probes 212020_s_at, 212021_s_at, 212022_s_at, and 212023_s_at) in unaffected skin and acne lesions from microarray datasets GSE53795 ($n = 12$ control skin, $n = 12$ acne skin; $p = 0.016$) and GSE6475 ($n = 12$ control skin, $n = 6$ acne skin; $p = 0.00002$). **c** Proliferation of ORS keratinocytes assessed as % confluence using an Incucyte video-microscope. Values at each time point were normalised to the first scan point of the Mock+Dox condition ($n = 4$/condition; $p = 0.0001$). **d** Number of colonies (normalised to the number of colonies in the corresponding Vehicle condition) and colony area of ORS keratinocytes overexpressing GATA6 ($n = 18$/condition; $p < 0.0001$ and $p = 0.037$, respectively). Representative dishes are shown. **e** Mock-infected or GATA6-infected sebocytes were treated with 1 μg ml$^{-1}$ Dox for 4 days. Representative histograms of propidium iodide (PI)-stained cells and quantification of % S phase cells compared to the Mock+Dox condition are shown ($n = 13$ for Mock conditions, $n = 11$ for GATA6 + Vehicle, $n = 14$ for GATA6+Dox; $p < 0.000001$). **f** Colony area of Mock or GATA6 SebE6E7 in the presence of 1 μg ml$^{-1}$ Dox ($n = 6$/condition; $p = 0.00004$). Representative dishes are shown. **g** Mock or GATA6 infected sebocytes treated with 1 μg ml$^{-1}$ Dox or vehicle for 4 days ($n = 6$ for Vehicle conditions, $n = 9$ for Dox conditions; $p = 0.799$). Representative images of TUNEL-stained cells and quantification of the percentage of TUNEL+ cells are shown. Negative (Neg Ctrl, $n = 5$) and positive (Pos Ctrl, $n = 6$) controls are shown. **a, g** Nuclei were counterstained with DAPI. Scale bars: 50 μm. **a–g** Data are presented as mean ± SD (**c**) or individual values with means (**b, d–g**) and were obtained from two (**a, f**), three (**d, g**), four (**c**), or five (**e**) independent experiments corresponding to *n* replicates. Statistical analyses were performed with two-tailed unpaired *t*-test (**b**), linear regression (**c**) or with ordinary one-way ANOVA (**d–g**). ns not significant; *$p$-value < 0.05; ***$p$-value < 0.0005; ****$p$-value < 0.00005. **b–g** Source data are provided as a Source Data file.

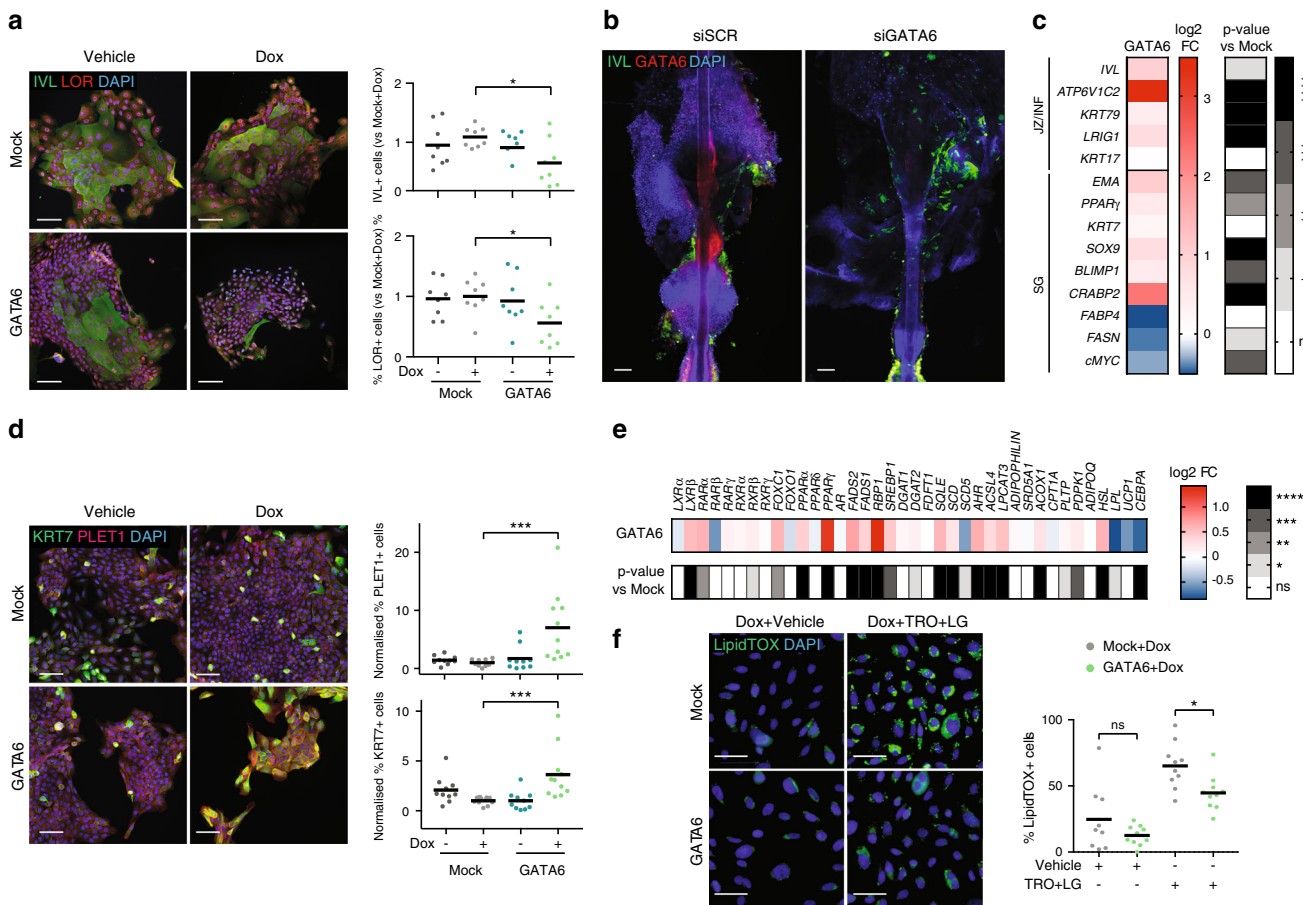

**Fig. 4 GATA6 regulation of ductal and sebaceous differentiation. a** Mock or GATA6-infected keratinocytes treated with 1 μg ml$^{-1}$ Dox for 5 days and immunolabelled for IVL and LOR ($n = 8$/condition; $p = 0.011$ and $p = 0.021$ respectively). Representative images and quantitation are shown. **b** Organ-cultured human HF transfected with siSCR or siGATA6 and maintained for 5 days. Representative images for GATA6 and IVL staining. **c–f** Mock or GATA6-infected SebE6E7 sebocytes treated with 1 μg ml$^{-1}$ Dox or vehicle for 4 (**c, e**) or 5 (**d**) days; 1 μg ml$^{-1}$ Dox and vehicle (DMSO) or 1 μM TRO + 0.1 μM LG for 4 days (**f**). **c, e** Gene expression in GATA6+Dox sebocytes is represented as log$_2$ FC versus Mock+Dox sebocytes ($n = 6$/condition in **c**, and $n = 20$/condition in **e**). **d** Representative images of PLET1 and KRT7 labelling and quantification are shown ($n = 10$ for Vehicle conditions and $n = 11$ for Dox conditions; $p = 0.0001$ and $p = 0.0002$, respectively). **f** Representative images of LipidTOX staining and quantification are shown ($n = 9$ for Mock+Dox +Vehicle and $n = 10$ for other conditions). **a–f** Scale bars: 50 μm (**f**), 100 μm (**a, b, d**). Nuclei were counterstained with DAPI (**a, b, d, f**). Data are represented as mean (**c, e**) or mean with individual values (**a, d, f**) and were obtained from two (**b, c**), three (**a, d, f**), or five (**e**) independent experiments corresponding to *n* replicates. Statistical analyses were performed with ordinary one-way ANOVA (**a, c, d, f**) or two-tailed multiple *t*-tests (**e**). ns not significant; *$p$-value < 0.05; **$p$-value < 0.005; ***$p$-value < 0.0005; ****$p$-value < 0.00005. **a, c–f** Source data are provided as a Source Data file.

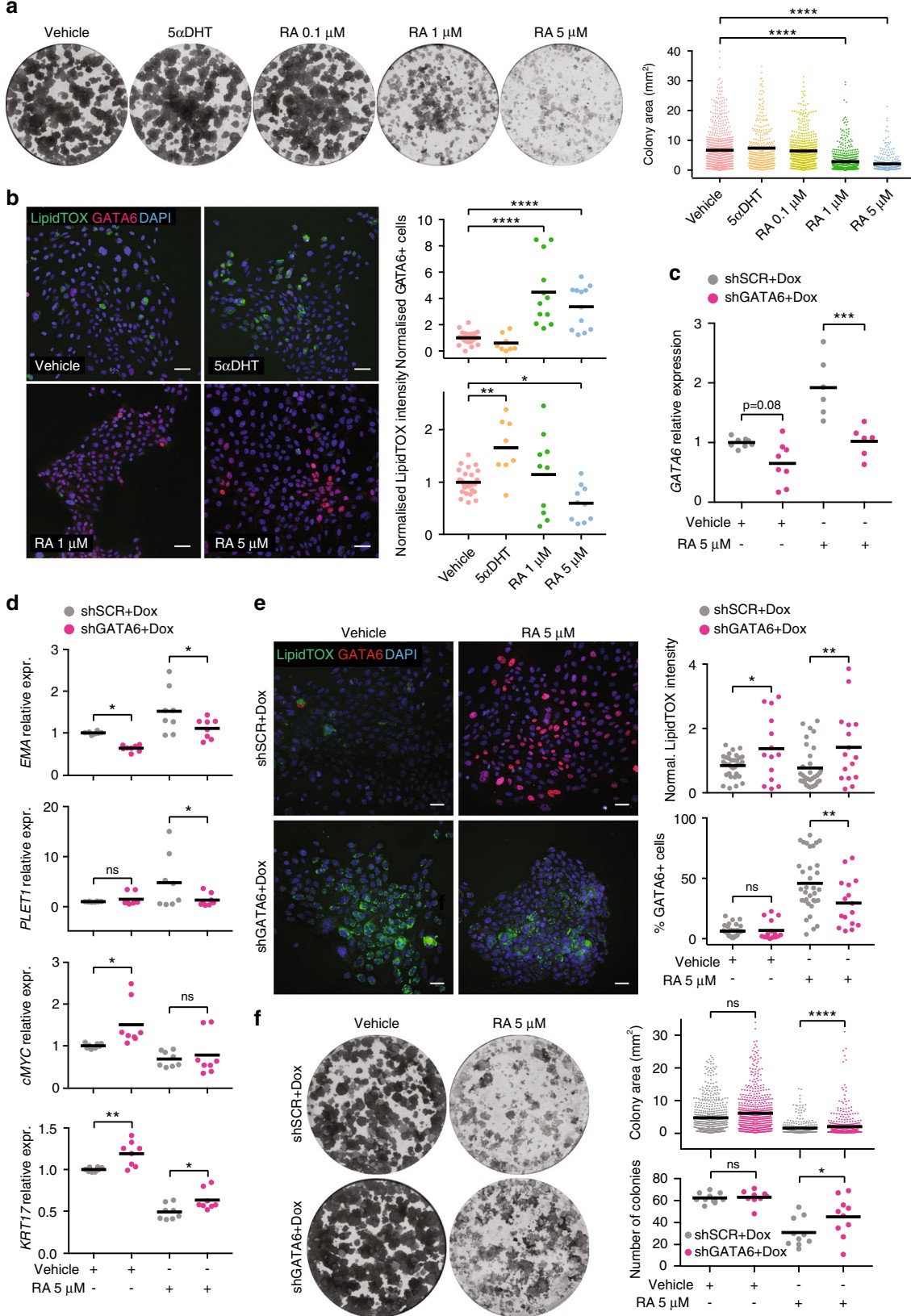

These data indicate a role for GATA6 in mediating the effects of RA on sebocyte proliferation, differentiation and lipid production.

**Role of GATA6 in immune regulation of the upper HF**. The upper pilosebaceous unit is an immunologically active region, where ductal keratinocytes and sebocytes express receptors that sense pathogen-associated molecular patterns (PAMP), such as toll-like receptors TLR2 and TLR4, secrete chemokines and anti-microbial peptides (AMP), and are able to interact with immune cells[5]. Activation of pro-inflammatory cascades is thought to

**Fig. 5 GATA6 overexpression phenocopies retinoic acid effect on sebocytes. a** Area of colonies formed by SebE6E7 sebocytes cultured with vehicle or 100 nM 5αDHT, 0.1 μM RA, 1 μM RA or 5 μM RA ($n = 10$ for Vehicle, $n = 6$ for 5αDHT, $n = 8$ for RA conditions). Representative dishes and quantitation are shown. **b** Sebocytes treated for 5 days with vehicle ($n = 26$) or 100 nM 5αDHT ($n = 8$), 1 μM RA or 5 μM RA ($n = 12$ for RA conditions for GATA6 analysis and $n = 10$ for RA conditions for LipidTOX analysis). Representative images of LipidTOX and GATA6 labelling and quantitation are shown. Values are expressed as FC versus vehicle condition. **c** Inhibition of *GATA6* expression by shGATA6 lentiviruses. Sebocytes were infected with shSCR or shGATA6 lentiviruses and treated with 1 μg ml$^{-1}$ Dox with vehicle or 5 μM RA for 5 days ($n = 8$ for vehicle conditions, $n = 6$ for RA conditions). RT-qPCR of *GATA6* expression normalised to housekeeping genes. Values are expressed as FC versus shSCR+Dox+Vehicle values. **d–f** Inhibition of GATA6 by shGATA6 reduces the effects of RA on sebocytes. Cells were treated as in **c**. **d** Expression of *EMA*, *PLET1*, *KRT17*, and *cMYC* ($n = 8$/condition). **e** Quantitation of representative images of LipidTOX and GATA6 labelling ($n = 34$ for shSCR conditions, $n = 14$ for shGATA6+Dox+Vehicle, $n = 16$ for shGATA6+Dox+ RA). Values are expressed as FC over shSCR+Dox+Vehicle condition. **f** Number of colonies and colony area ($n = 10$/condition). Representative dishes are shown. **b**, **e** Nuclei were counterstained with DAPI. Scale bars: 50 μm (**b**, **e**). **a–f** Data are means with individual values and were obtained from three (**a**, **b**, **e**) or four (**c**, **d**, **f**) independent experiments corresponding to $n$ replicates. Statistical analysis was performed with ordinary one-way ANOVA (**a–f**). ns not significant; *$p$-value < 0.05; **$p$-value < 0.005; ***$p$-value < 0.0005; ****$p$-value < 0.00005. **a–f** Source data are provided as a Source Data file.

contribute to the pathogenesis of acne[3,13,14], and could be involved in the very early stages of the disease[42]. We therefore evaluated the expression of chemokines and AMP in SebE6E7 sebocytes in response to altered GATA6 expression and bacterial challenge (Fig. 6a).

Inhibiting GATA6 with shRNA led to a significant upregulation of *IL8* and β-defensin 1 (*βDEF1*) and a decrease in *IL6* and cathelicidin (*CAMP*). Conversely, GATA6 overexpression triggered an increase in *IL6*, *IL10*, *βDEF4*, *TLR2*, *TLR4* and a decrease in *IL17*. There was also a trend towards decreased *IL8* expression (Fig. 6a).

To mimic the effect of *P. Acnes* on sebocytes, we treated control or GATA6-expressing sebocytes with peptidoglycan (PGN), which is the main component of the Gram+ bacterial cell wall and is recognised by TLR2[43]. We found that GATA6 expression led to a decrease in PGN-induced *IL8*, *IL17*, *S100A7*, as well as to an increase in dermcidin (*DCD*) expression (Fig. 6a).

We then explored the effect of live *P. acnes* bacteria on control and GATA6 expressing sebocytes. GATA6 expression triggered an increase in *IL6*, *TSLP*, *TLR2*, *TLR4*, and *βDEF4* expression relative to controls, whereas there was a strong decrease in *S100A7* and a trend towards a decrease in *IL17* expression (Fig. 6a). Intriguingly, expression of GATA6 alone or in the presence of a bacterial insult drove a slight increase of *IFNγ* (Fig. 6a).

To determine whether the transcriptional changes correlated with changes in protein levels, we probed a cytokine array with lysates of control or GATA6+ SebE6E7 cells treated with vehicle or PGN (Fig. 6b). Most cytokines were expressed at low levels and the main effect of GATA6 induction was IL10 upregulation in the presence or absence of PGN. Treatment with PGN was required for an increase in IFNγ and SDF-1/CXCL12 expression upon GATA6 induction (Fig. 6b).

To evaluate the ability of sebocytes to communicate with immune cells, we tested the effect of GATA6 expression on unstimulated or IFNγ-stimulated SebE6E7 cells. Independent of IFNγ treatment, GATA6+ sebocytes exhibited an increase in expression of PD-L1, the main ligand for the immunosuppressive receptor PD1[44]. HLA-DR was downregulated in IFNγ-stimulated GATA6-expressing sebocytes, while slightly upregulated at steady state (Fig. 6c). CD40 expression was downregulated in unstimulated GATA6+ cells, yet this result was not statistically significant. CD80 and CD86, which can be both co-activating or co-inhibitory molecules, were expressed at low levels, but upregulated upon GATA6 induction and IFNγ treatment (Supplementary Fig. 7a). Finally, we found that *GATA6* expression was strongly upregulated by PGN or *P. acnes* treatment (Supplementary Fig. 7b, c).

These results suggest that GATA6 modulates the immune repertoire of sebocytes by inducing the expression of anti-inflammatory molecules, in particular IL10 and PD-L1, and downregulating acne-driven *IL8* and *IL17* cytokines. However,

GATA6 is also responsible for the induction of low levels of pro-inflammatory molecules such as *IL6*, *TLR2*, *TLR4*, and *IFNγ*. Therefore, we postulate that GATA6 contributes to immune regulation of pilosebaceous units and anti-microbial responses[45].

**The TGFβ pathway is activated by GATA6 to silence IFE fate.** Although human sebaceous cell lines in 2D culture have been widely used to study sebaceous gland biology, major limitations exist, such as the failure to recapitulate the architecture of the gland[28]. For this reason, we developed a 3D sebaceous organoid model using SebE6E7 sebocytes to better recapitulate SG homeostasis. Cells were seeded in 3D discs of Matrigel in a basal medium supplemented with EGF and bovine pituitary extract. After 24 h, we observed structures comprising 2–3 cells (Fig. 7a). After a week in culture, solid organoids had formed. Immunostaining revealed that 1-week sebaceous organoids were organised such that cells at the periphery expressed the ductal marker PLET1 and cells in the centre expressed the lipid enzyme FASN (Fig. 7a). Thus, the organoids recapitulated the basic aspects of SG organisation.

We next compared the effects of RA and RepSox (a TGFβ inhibitor that targets ALK5) since RA and TGFβ have synergistic actions on epidermal keratinisation[46]. RA-treated organoids were smaller and RepSox-treated organoids were larger than vehicle-treated organoids (Fig. 7b). Gene expression profiling revealed a strong induction of IFN/IFE differentiation genes, such as *IVL*, *TGM1*, or *SBSN*, upon RepSox treatment. In contrast, RA treatment downregulated the IFE differentiation programme while upregulating several ductal and sebaceous markers (Fig. 7c).

We next analysed sebaceous organoids formed by GATA6-overexpressing or shGATA6-overexpressing SebE6E7 cells and controls (Mock-infected or shSCR-infected cells). As predicted, GATA6+ sebaceous organoids had a similar gene expression profile to RA-treated organoids. In contrast, shGATA6+ sebaceous organoids resembled RepSox-treated organoids, suggesting that inhibition of GATA6 phenocopied inhibition of TGFβ signalling (Fig. 7d). To test this, we overexpressed GATA6 or shGATA6 in RepSox-treated organoids. Compared to RepSox-treated control organoids, GATA6 expression led to a down-regulation of the IFE gene signature and increased expression of ductal and sebaceous markers. In contrast, inhibition of GATA6 further enhanced expression of genes associated with the IFE fate (Fig. 7e). In agreement with the RA effect on organoid size, GATA6 organoids were smaller than controls and did not increase in size when treated with RepSox (Supplementary Fig. 8a). There was a trend towards larger organoids upon GATA6 knockdown, although this was not statistically significant (Supplementary Fig. 8a).

Immunostaining of sebaceous organoids for PLET1 revealed similarities between vehicle-treated or RepSox-treated

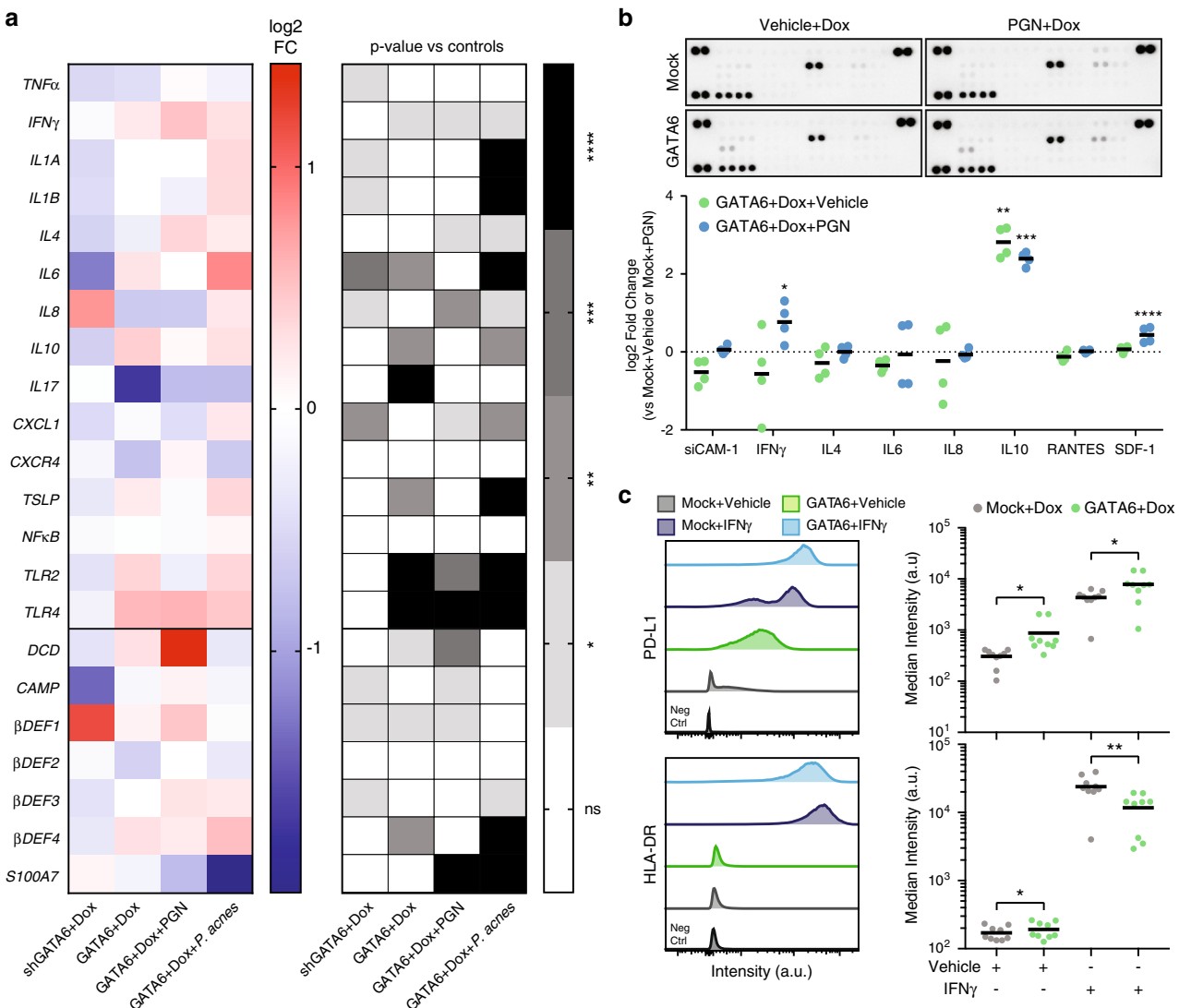

**Fig. 6 Effects of GATA6 on immunological repertoire of sebocytes. a** SebE6E7 cells were infected with shGATA6 or shSCR (column 1, *n* = 12/condition), or with GATA6 or Mock (columns 2–4) lentiviruses and treated with 1 μg ml⁻¹ Dox or vehicle for 3–4 days (*n* = 32/condition for column 2, *n* = 16/ condition for column 3, *n* = 48/condition for column 4). Sebocytes were co-treated with 1 μg ml⁻¹ PGN for 3 days (column 3) or *P. acnes* for 16 h (column 4). Expression of cytokines and AMP was assessed by RT-qPCR and normalised to housekeeping gene expression. Values are expressed as log₂ FC versus corresponding Mock or shSCR values. **b** Cytokine protein expression was assessed using cytokine arrays in Mock or GATA6-infected sebocytes treated with Vehicle+Dox or Dox+PGN (*n* = 4/condition). Representative arrays are shown. Values are expressed as log₂ FC versus Mock+Dox+Vehicle values for GATA6+Dox+Vehicle condition, or versus Mock+Dox+PGN for GATA6+Dox+PGN values. **c** Mock and GATA6-overexpressing sebocytes were treated with 1 μg ml⁻¹ Dox and 1 U ml⁻¹ human recombinant IFNγ or vehicle for 3 days. Median intensity of PD-L1 and HLA-DR staining was analysed by flow cytometry (*n* = 9/condition). Representative FACS plots are shown. **a–c** Data are mean (**a**) or mean ± SD (**b**) or mean with individual values (**c**) and were obtained from two (**b**) or at least four (**a**, **c**) independent experiments corresponding to *n* replicates. Statistical analyses were performed with two-tailed multiple *t*-tests (**a**) or ordinary one-way ANOVA (**b**, **c**). ns not significant; *$p$-value < 0.05; **$p$-value < 0.005; ***$p$-value < 0.0005; ****$p$-value < 0.00005. **a–c** Source data are provided as a Source Data file.

shGATA6-expressing organoids and RepSox-treated control organoids. Organoids in these three conditions displayed a disorganised architecture and membrane ruffling (Fig. 7f). While GATA6+ organoids expressed more PLET1 and KRT7, as expected, vehicle-treated or RepSox-treated shGATA6+ or RepSox-treated control organoids showed a substantial increase in IVL expression, indicative of IFE fate induction (Fig. 7f, g). We observed an increase in SMAD2/3 phosphorylation in GATA6-expressing sebocytes in 2D culture (Supplementary Fig. 8b), as well as in organoid cultures (Fig. 7h). Moreover, in healthy human skin P-SMAD2/3, indicative of TGFβ activation,

was found in the suprabasal cells of the INF (Fig. 7i) where GATA6 is expressed.

By analysing single-cell RNA sequencing data from human scalp epithelial cells[29], we obtained DEG for *GATA6*+ cells as compared to all other epidermal and adnexal cells present in the dataset (Supplementary Data 2). We used the IPA Upstream Regulator Analysis tool to interrogate the transcriptomics data from *GATA6*+ cells. This analysis revealed that several components of the TGFβ pathway were able to resolve the gene expression changes in *GATA6*+ cells compared to *GATA6*– cells (Fig. 7j and Supplementary Fig. 8c). This analysis also identified FoxO signalling

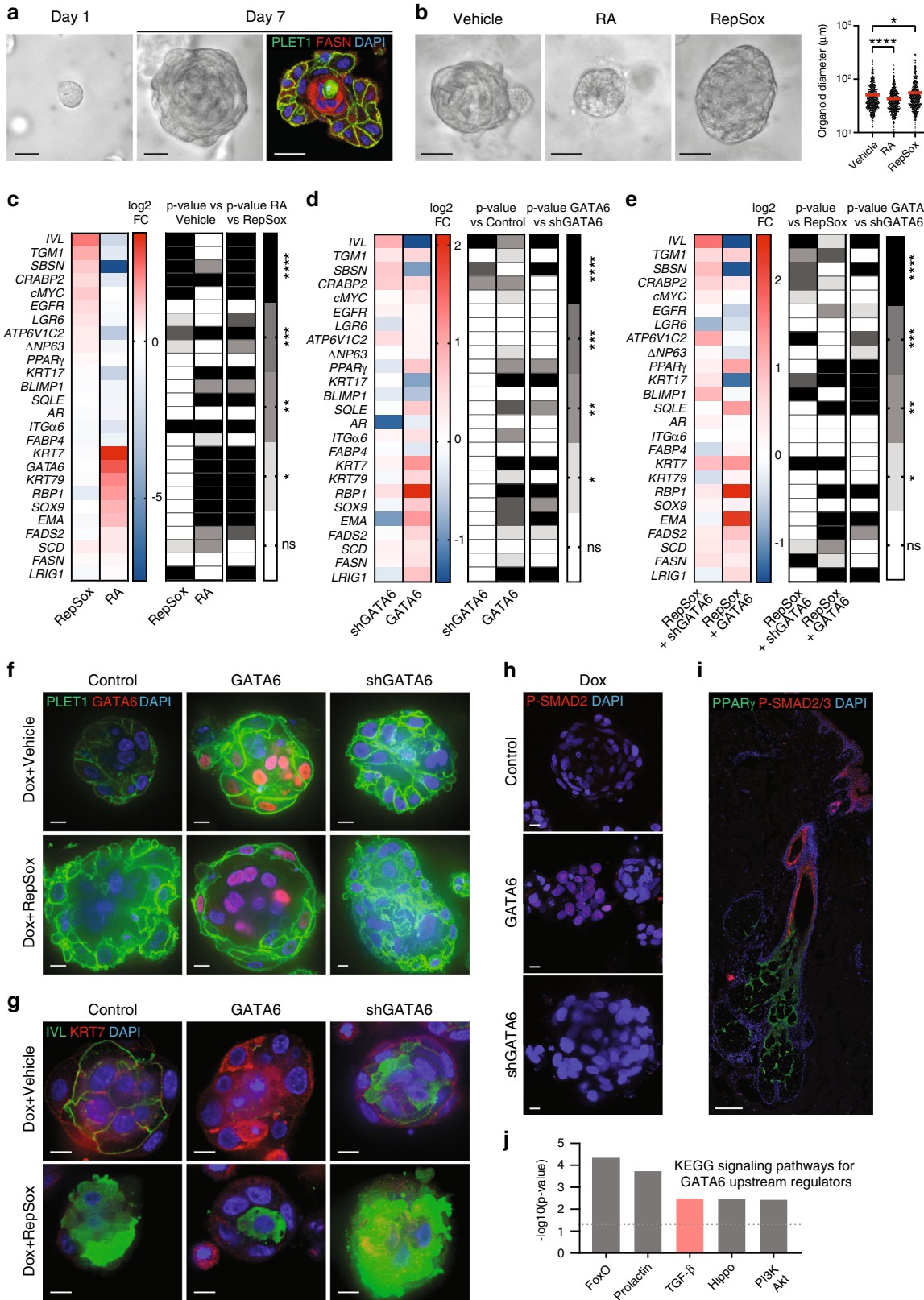

(Fig. 7j), consistent with the pivotal role of FoxO1 in acne[12]. In addition, analysis of microarray data revealed that TGFβ signalling was downregulated in acne (Supplementary Fig. 8d, e).

In summary, our work identifies GATA6 as a regulator of the human upper pilosebaceous unit in humans. Its loss contributes to the pathogenic features of acne, while its induction by RA may play a role in the resolution of acne (Fig. 8). By developing a sebaceous organoid model, we obtained evidence that GATA6-mediated TGFβ activation is a key process controlling the repression of IFE fate and promoting JZ/SD differentiation.

**Fig. 7 GATA6 activates TGFβ pathway to repress IFE fate. a, b** Bright-field micrographs of SebE6E7 organoids. Sebaceous organoids were labelled for PLET1 and FASN in **a**. Organoids in **b** were treated with vehicle (DMSO), 5 μM RA or 25 μM RepSox for 6 days ($n = 110$ for Vehicle condition, $n = 98$ for RA and RepSox conditions). Representative images and quantification of organoid diameter are shown. **c** Sebaceous organoids were grown as in **b** and analysed by RT-qPCR. Values were normalised to housekeeping genes and presented as $\log_2$ FC versus vehicle condition ($n = 24$ for RepSox condition, $n = 26$ for RA condition). **d** Sebaceous organoids formed by mock-, GATA6-expressing, shSCR-expressing or shGATA6-expressing SebE6E7 sebocytes were grown for 7 days in the presence of 1 μg ml$^{-1}$ Dox. Analysis was performed as in **c** and values are represented as $\log_2$ FC versus Mock or shSCR conditions ($n = 10$/condition). **e** Sebaceous organoids were grown as in **d** in the presence of 25 μM RepSox. Analysis was performed as in **d** and values are represented as $\log_2$ FC versus Mock+Dox+RepSox or shSCR+Dox+RepSox conditions ($n = 10$/condition). **f, g** Cells were treated as in **d, e**. Representative images for GATA6 and PLET1 (**f**) or IVL and KRT7 (**g**) staining. **h** GATA6, shGATA6, or control sebaceous organoids were grown for 7 days in the presence of 1 μg ml$^{-1}$ Dox and stained for P-SMAD2. **i** Back skin of 60-year-old male labelled for PPARγ and P-SMAD2/3. **j** KEGG signalling pathways analysis for GATA6 upstream regulators displayed in Supplementary Fig. 8c. **a–i** Scale bars: 10 μm (**f–h**), 25 μm (**a, b**) and 200 μm (**i**). Nuclei were counterstained with DAPI (**a, f–i**). Data are mean (**c–e**), or mean with individual values (**b**), and were obtained from two (**a, f–h**), three (**d, e**) or five (**b, c**) independent experiments corresponding to $n$ replicates. Statistical analyses were performed with ordinary one-way ANOVA (**b–e**). ns not significant; *$p$-value < 0.05; **$p$-value < 0.005; ***$p$-value < 0.0005; ****$p$-value < 0.00005. **b–e** Source data are provided as a Source Data file.

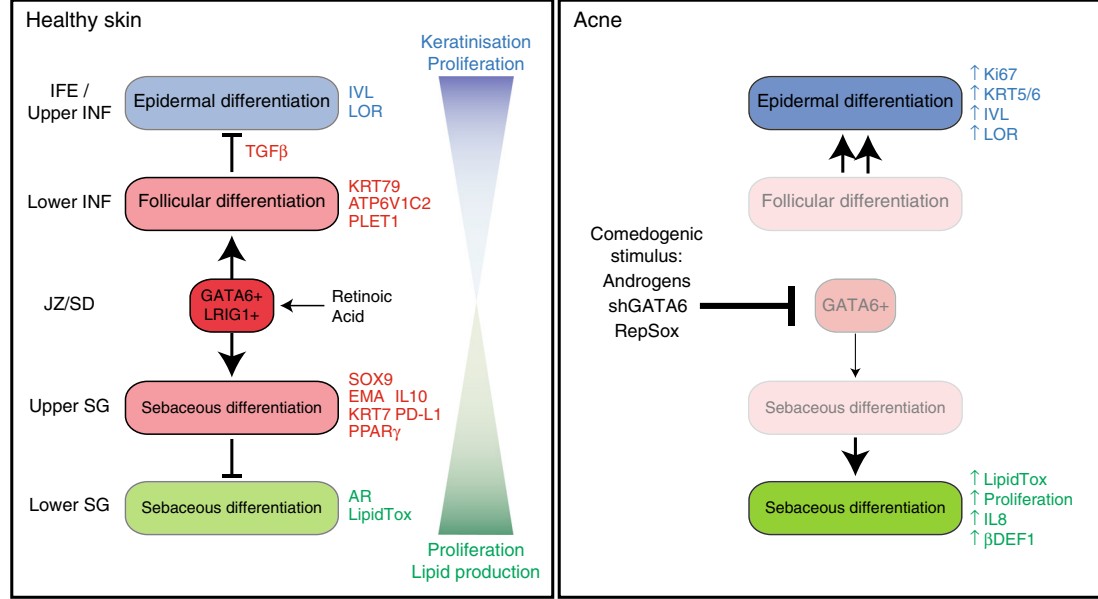

**Fig. 8 Model for GATA6 role in the human upper pilosebaceous unit.** GATA6 marks a population of cells of the human upper pilosebaceous unit that contribute both to the follicular (lower INF/JZ/SD) and sebaceous (upper SG) compartments. GATA6 limits the proliferation and the extent of differentiation within these two compartments. In addition, it negatively regulates IFE/upper INF fate through TGFβ signalling induction. During comedogenesis, GATA6+ progenitors are targeted and GATA6 expression is reduced. This causes a switch in lineage determination within the upper pilosebaceous unit that favours the IFE/upper INF fate.

## Discussion

Our work provides evidence that GATA6 controls several physiological processes contributing to homoeostasis of the upper pilosebaceous unit in human skin, and that its expression is markedly reduced in acne skin. Our findings suggest that GATA6 triggers ductal and sebaceous differentiation and limits, in parallel, cell proliferation and lipid production to prevent hyperseborrhoea. GATA6 also orients sebocytes towards an antiinflammatory phenotype. Lastly, by repressing IFE fate through activation of the TGFβ pathway, GATA6 may be preventing the formation of comedones (Fig. 8).

Acne occurs most frequently in adolescents, and a clear epidemiological link exists between androgens and acne[3]. While oestrogens confer a protection against acne, as demonstrated by the excellent therapeutic effect of some combined oral contraceptive pills and hormonal antiandrogen treatments[47], patients with acne may have clinical signs of hyperandrogenism and, more rarely, biological hyperandrogenism[48]. Acne also occurs in a majority of women suffering from polycystic ovary syndrome, which is linked to an excess of androgens[49]. In addition, 5αDHT, the peripheral metabolite of testosterone, has been shown to increase sebocyte proliferation and lipid production by binding to androgen receptors present in sebocytes[50].

While GATA6 expression is restricted to the upper SG, AR is expressed in the bottom of the gland in mouse and human skin (Supplementary Fig. 1f and see ref. [23]) We previously showed that GATA6 directly represses *AR* gene expression in mouse skin and that *AR* is induced upon *GATA6* knockdown[23]. Therefore, a negative auto-regulatory loop exists between GATA6 and AR, defining two distinct compartments in the SG. At puberty onset, upregulation of AR signalling is likely to explain the downregulation of GATA6 we observe in acne samples (Fig. 2 and Supplementary Fig. 2).

GATA6 controls a network of genes involved in INF/JZ and SG homoeostasis, such as *KRT79* and *PPARγ* in mouse and human skin (Fig. 4c, see ref. [23,51]). Therefore, one would predict that GATA6 downstream targets would be silenced in acne since GATA6 is reduced. Indeed, expression of KRT79 was lost in five out of seven comedone samples from acne patients[21]. In contrast, expression of *KRT17*, another INF keratin that was negatively regulated by GATA6 in our experiments (Fig. 4c), is maintained in acne comedones[21]. Involvement of PPARγ in acne pathogenesis

has been hypothesised, given its inducing role in sebogenesis[30]. However, PPARγ and two of its target genes were reduced in acne SG as compared to healthy skin[52]. These results support our finding of downregulation of GATA6 during acne pathogenesis.

We previously identified GATA6 as a marker of sebaceous differentiation in benign and malignant mouse and human sebaceous tumours. In addition, *GATA6* knockdown in DMBA-treated K14ΔNLef1 mice led to an increased rate of tumour formation and increased the number of tumours per mouse. This led us to postulate that GATA6 is a tumour suppressor during sebaceous carcinogenesis, likely through controlling DNA mismatch repair response mechanisms[24]. As also suggested by missense mutations in the GATA6 gene in human sebaceous tumours[24], we propose that chronic GATA6 loss may be associated with tumour progression in combination with mutations in other genes. However, there is no evidence that GATA6 loss per se is directly involved in skin tumour initiation, consistent with the absence of tumour comorbidity in acne.

Sebocytes and ductal keratinocytes are actively involved in immune signalling of the upper pilosebaceous unit[53–55]. As we have demonstrated (Fig. 6), GATA6 increases components of an active innate immune response, such as *TLR2*, *TLR4*, *IL6*, *IFNγ*, and *AMP*, increases expression of immunosuppressive molecules such as IL10 and PD-L1 and reduces expression of *IL17* and *IL8*, key mediators of the exacerbated immune response in acne[30,31].

Recent evidence suggests that the immune response to *P. acnes* may be more complex than previously envisioned[13], and in the future it will be interesting to explore how GATA6-expressing cells interact with other components of the immune system. Activation of immune signalling pathways can be a double-edged sword, necessary to clear pathogens but harmful when too sustained or uncontrolled. A recent report highlights that persistence of inflammation in acne lesions is linked to prolonged lesions, scar formation and loss of SG[56]. This suggests that immune overactivation is detrimental in acne and that anti-inflammatory treatments are needed[57].

While GATA6 positively regulates ductal and sebaceous fates, it is also an inhibitor of IFE fate. This may be crucial for decreasing proliferation and increasing keratinisation within the IFN, thereby preventing the occlusion of HF pores. In the vicinity of acne lesions in which GATA6 was lost, we observed thicker and more proliferative IFE (Fig. 2a and Supplementary Fig. 2), reinforcing the hypothesis that GATA6 negatively controls IFE fate. We identified TGFβ activation as a key mechanism in this process (Fig. 7 and Supplementary Fig. 8) by treating sebaceous organoids with RepSox, a TGFβRI inhibitor that blocks Activin, BMP and TGFβ, the three main pathways of the TGFβ superfamily[58], and also by analysing single cell RNA transcriptomics of *GATA6+* cells in human scalp samples[29].

The role of the TGFβ superfamily in controlling the behaviour of HF stem cells[59] and dermal fibroblasts[60] has been extensively analysed. However, few studies have considered a possible role for TGFβ signalling in the upper pilosebaceous unit. Reversible inhibition of TGFβ signalling in TGFβ3-expressing cells (HF lineages and epidermal suprabasal cells) of mouse skin leads to HF defects, together with aberrant epithelial differentiation in the upper HF, resulting in infundibular cysts expressing IVL and FLG and containing Oil Red O-stained lipids. Moreover, in this model the proliferation of isthmus keratinocytes was increased and expression of LRIG1 was lost[61]. A mouse knock-out of *SMAD4* in KRT5+ cells led to enlarged INF and dermal keratotic cysts[62].

Several reports indicate that loss of TGFβ triggers keratinocyte hyperproliferation and can participate in tumour initiation[63–65]. This is in agreement with our findings that GATA6 downregulates the IFE differentiation programme and proliferation through TGFβ activation. RA induces TGFβ2 in the dermal papilla, which could explain the hair loss associated with retinoid treatment[66].

A genome-wide association study comparing severe cases of acne with healthy individuals identified three new susceptibility loci containing genes involved in the TGFβ pathway[67]. Expression of two of these genes was significantly reduced in acne samples[67], validating our own observation of a downregulation of the TGFβ pathway in acne. We showed that GATA6 expression led to increased P-SMAD2/3 (Fig. 7h and Supplementary Fig. 8b), which transduces signalling by TGFβ, activin and nodal ligands[58]. Consistent with this, acne is one of the main side effects of treatment with bimagrumab, a monoclonal antibody targeting the activin receptor ACVRIIB[68].

In conclusion, our work identifies GATA6 as a regulator of the upper pilosebaceous unit in human skin. Its loss is likely to contribute to the pathogenic features of acne, opening up new avenues for acne research and treatment.

## Methods

**Human tissue**. All human tissue samples were collected after informed consent and processed in compliance with all relevant ethical regulation, including the Declaration of Helsinki's recommendations on human research and the UK Human Tissue Act (2004).

Samples of adult healthy skin (surplus surgical waste) were obtained from patients undergoing plastic surgery. Collection for research use of adult healthy skin was approved by the National Research Ethics Service (UK) (Human Tissue Authority Licence No. 12121, Research Ethics Committee No. 14/NS/1073).

To isolate the foetal primary outer root sheath (ORS) keratinocytes, foetal back skin was obtained with appropriate ethical approval from the UK Human Developmental Biology Resource (www.hdbr.org).

Acne vulgaris samples were obtained from diagnostic biopsies retrieved from the pathology department archives of Hôpitaux Universitaires Paris Nord Val de Seine following approval by the departmental committee (19BX00114). For all patients, the study was performed after establishing the absence of registered opposition to the use of tissue samples according to the guidelines of the French Bioethics Law for retrospective noninterventional research studies. Biopsies were obtained by dermatologists (none of the co-authors) from Hôpitaux Universitaires Paris Nord Val de Seine. For all cases, there was enough formalin-fixed paraffin embedded material remaining after the diagnosis had been established to qualify the samples for additional immunohistochemical studies. Thus diagnostic surplus tissue was used for research.

**Cell culture**. Primary human keratinocytes (strain km) were isolated from IFE (neonatal foreskin) and used at passages 4–8. ORS keratinocytes (strain k12026) were isolated from hair follicles (foetal back skin) and used at passages 3–5. The SebE6E7 sebocyte cell line was generated from sebaceous glands micro-dissected from adult human facial skin and immortalised by retroviral transduction of *HPV16/E6E7* genes[32]. 3T3-J2 cells were a gift from James Rheinwald and can be purchased from Kerafast (EF3003). Keratinocytes and SebE6E7 sebocyte stocks were maintained on mitotically inactivated 3T3-J2 cells in complete FAD medium, comprising one part Ham's F12, three parts Dulbecco's modified Eagle medium (DMEM) and $10^{-4}$ M adenine (Gibco), supplemented with 10% foetal bovine serum (Gibco), 1% penicillin/streptomycin (Gibco), 1% L-glutamine (Gibco), 1.8 mM calcium chloride (Sigma-Aldrich), 10 ng ml$^{-1}$ EGF (Peprotech), 0.5 μg ml$^{-1}$ hydrocortisone (Sigma-Aldrich), 5 μg ml$^{-1}$ insulin (Sigma-Aldrich) and $10^{-10}$ M cholera enterotoxin (Sigma-Aldrich). 3T3-J2 cells were cultured in high-glucose DMEM (Sigma-Aldrich) with 10% adult bovine serum (Thermo Fisher Scientific) and 1% penicillin/streptomycin (Gibco), and were mitotically inactivated by treating for 2 h with 4 μg ml$^{-1}$ mitomycin (Sigma-Aldrich). HEK-293 cells were purchased from ATCC (CRL-1573) and cultured in high-glucose DMEM with 10% adult bovine serum and 1% penicillin/streptomycin (Gibco). All cell stocks were routinely tested for mycoplasma contamination and were negative.

Unless specified otherwise, experiments on keratinocytes and SebE6E7 sebocytes were performed in the absence of feeders. Cells were seeded on plates coated with 25 μg ml$^{-1}$ rat-tail collagen type I in PBS (BD Biosciences) and cultivated in keratinocyte serum-free medium (KSFM) supplemented with 30 μg ml$^{-1}$ bovine pituitary extract and 0.15 ng ml$^{-1}$ EGF (Thermo Fisher Scientific).

Live-cell video-microscopy was performed using an IncuCyte Zoom (Essen BioScience) device placed in a temperature and $CO_2$-controlled incubator.

For serum-induced differentiation, keratinocytes were grown to confluence in KSFM and then switched to KSFM supplemented with 10% foetal bovine serum.

Lentivirus-infected keratinocytes were treated with vehicle or 1 μg ml$^{-1}$ Dox for 16 h before starting each assay.

Clonogenicity assays were performed using 1000 cells plated on mitotically inactivated 3T3-J2 cells per well of six-well plates. After 12 days, feeders were removed. Colonies were fixed in 4% paraformaldehyde (Sigma-Aldrich) for 10 min

then stained with 1% Rhodanile Blue (1:1 mixture of Rhodamine B and Nile Blue chloride) (Sigma-Aldrich).

Two *P. acnes* strains were cultivated under anaerobic conditions in liquid reinforced clostridial medium (RCM) at 37 °C for 5 days. Sebocytes were stimulated overnight with *P. acnes* suspensions diluted in KSFM medium at optical densities (OD) of 0.01, 0.1 or 1 as measured by the absorbance read at 620 nm.

**Chemicals**. To induce sebaceous differentiation, SebE6E7 sebocytes were treated with 1 μM troglitazone (TRO, PPARγ agonist) (Sigma-Aldrich) with or without 0.1 μM LG100268 (LG, RXR agonist) (Sigma-Aldrich) for 4 days. Where indicated, SebE6E7 sebocytes were treated with 100 nM 5α-dihydrotestosterone (5αDHT) (Sigma-Aldrich) and/or 0.1, 1 or 5 μM all-trans retinoic acid (RA) (Sigma-Aldrich) for 5 (RT-qPCR and stainings) to 12 (colony formation assay) days. For immunological assays, cells were stimulated with 1 μg ml$^{-1}$ Peptidoglycan (PGN) from Staphylococcus aureus (Sigma-Aldrich) for 72 h or 1 U ml$^{-1}$ human recombinant IFNγ (ThermoFisher Scientific) for 72 h. Sebaceous organoids were treated with 25 μM RepSox (TGFβ type I receptor/ALK5 inhibitor) (R&D Systems) or 5 μM RA for 6 days.

All chemicals were dissolved in sterile DMSO, except 5αDHT which was dissolved in methanol, and PGN and IFNγ in sterile water. The same concentration of diluent was used as the Vehicle control condition.

**siRNA transfection of microdissected pilosebaceous units**. To obtain human whole pilosebaceous units[69], human skin was digested with Dispase (Sigma-Aldrich) overnight at 4 °C to detach the epidermis from the dermis. Epidermal sheets were subsequently microdissected to isolate individual intact pilosebaceous units.

A pool of ON-TARGETplus siRNAs (Dharmacon/Horizon Discovery) was used for gene knockdowns. Non-targeting (siSCR) or *GATA6*-targeting siRNAs (siGATA6) pools were a mix of four sets of RNAi oligos (siSCR: UGGUUUACAU GUCGACUAA, UGGUUUACAUGUUGUGUGA, UGGUUUACAUGUUUUCU GA, UGGUUUACAUGUUUUCCUA; siGATA6: AAGACUUGCUCUGGUAA UA, GAACAGCGAGCUCAAGUAU, GCAGAAACGCCGAGGGUGA, GGUGA UGACUGGUGCGGGA). Transfection of siRNAs was performed in accordance with the transfection reagent's manufacturer's instructions. Non-targeting (siSCR) or *GATA6*-targeting siRNAs (siGATA6) were diluted in Williams E medium (Sigma-Aldrich) supplemented with 2 mmol l$^{-1}$ L-glutamine, 10 μg ml$^{-1}$ insulin, 10 ng ml$^{-1}$ hydrocortisone and 100 U ml$^{-1}$ penicillin/streptomycin, in a 12-well plate. siRNAs were incubated for 15 min with 4 μl of INTERFERin transfection reagent (PolyPlus Transfection). 10–20 pilosebaceous units were then placed in every well with 1 ml Williams E medium. The final volume was 1.2 ml per well and the final siRNA concentration 30 nM. The medium was changed to fresh complete Williams E medium 16 h after transfection. After 72 h some of the pilosebaceous units were collected and lysed for RNA extraction using the RNeasy Mini kit (Qiagen) according to manufacturer's instructions for downstream analysis to assess knockdown efficiency. One hundred and twenty hours of post-transfection, the remaining pilosebaceous units were fixed with 4% paraformaldehyde for 10 min at room temperature, washed with PBS (Sigma-Aldrich), and processed for staining and immunofluorescence microscopy.

**Sebaceous organoids**. Dissociated SebE6E7 sebocytes were resuspended in ice-cold Growth Factor Reduced Matrigel (Corning) at a density of 0.1 million cells per 25 μl in one well of a 24-well plate. Droplets of Matrigel and cells were seeded at the centre of each well, preventing the drop from touching the well edges. Plates were incubated for 30 min at 37 °C to obtain Matrigel solidification. Five hundred microliters of KSFM medium containing 30 μg ml$^{-1}$ bovine pituitary extract and 0.15 ng ml$^{-1}$ EGF were added per well. The next day, treatments were started for 6 days, adding fresh reagent every 2 days. After 7 days in culture, organoids were harvested for analysis.

**Plasmids**. Human *GATA6* open reading frame (ORF) (NCBI reference sequence NM_005257.5) cloned into a pReceiver-M02 expression plasmid was purchased from GeneCopoeia (Tebu-Bio). *GATA6* ORF was subcloned into the pCW57-GFP-2A-MCS lentiviral vector (gift from Adam Karpf; Addgene plasmid # 71783). Tet-pLKO-puro-Scrambled was a gift from Charles Rudin (Addgene plasmid # 47541)[70], and tet-pLKO-shGATA6-puro from Kevin Janes (Addgene plasmid # 72615)[71]. psPAX2 and pMD2.g packaging plasmids were gifts from Didier Trono (Addgene plasmid # 12260 and # 12259, respectively).

**Lentiviral production and infection**. HEK-293 cells were transfected with a lentiviral vector and psPAX2 and pMD2.g packaging plasmids (ratio 5.5:5:4:1) using JetPrime transfection reagent (Polyplus transfection) following the manufacturer's instructions. Supernatants were collected 48 and 72 h after transfection, passed through a 0.45 μm filter, concentrated using Lenti-X concentrator (Clontech) and centrifugated at 1500×*g* for 45 min at 4 °C. pCW57-GFP-2A-GATA6 plasmid was used to produce GATA6 lentiviruses, while pCW57-GFP-2A-MCS was used for Mock lentiviruses. Tet-pLKO-puro-Scrambled plasmid was used to produce shSCR lentiviruses and tet-pLKO shGATA6 for shGATA6 lentiviruses.

Cells were infected with appropriate amounts of lentiviruses in KSFM containing 5 μg ml$^{-1}$ polybrene (EMD Millipore). Two days post-infection, cells were subjected to 2 μg ml$^{-1}$ Puromycin (Thermo Fisher Scientific) selection for 3 days. Cell stocks were maintained without Doxycycline (Sigma-Aldrich). Unless stated otherwise, 1 μg ml$^{-1}$ Doxycycline (diluted in sterile water) was added to the growth medium to induce expression of the transgene during experiments.

**Immunofluorescence microscopy**. To analyse tissues, skin biopsies or reconstituted skin samples were either freshly embedded in optimal cutting temperature (OCT) compound (Life Technologies) and stored at −80 °C or were fixed in 10% neutral buffered formalin solution (Sigma-Aldrich) at 4 °C for 24 h and embedded in paraffin wax (Thermo Fisher Scientific) after dehydration. Ten micrometer OCT sections were cut using a Thermo Cryostar Nx70 (Thermo Fisher Scientific) and fixed for 10 min in 4% paraformaldehyde. Eight micrometer paraffin sections were cut using a Microme HM355s (Thermo Fisher Scientific), heated at 60 °C overnight, deparaffinised and subjected to 20 min heat-mediated antigen retrieval in 10 mM Sodium Citrate buffer, pH 6.0 (Sigma-Aldrich).

OCT or paraffin sections were permeabilised and blocked for 1 h in freshly prepared PB buffer (0.5% skim milk powder, 0.25% gelatin from cold water fish skin, 0.5% Triton X-100, 20 mM HEPES, 0.9% NaCl, pH 7.2 (all reagents from Sigma-Aldrich)). Primary antibodies were diluted in PB buffer (or in 10% donkey serum (Sigma-Aldrich) when labelling phosphoproteins)) and incubated overnight at 4 °C. Sections were washed with PBS and then labelled with secondary antibodies and 1 μg ml$^{-1}$ DAPI (Thermo Fisher Scientific) for 1 h at room temperature, washed with PBS, and mounted with Prolong Gold Antifade Mountant (Thermo Fisher Scientific).

Cells for labelling were grown on coverslips or in 96-well glass-bottom microplates and fixed for 10 min in 4% paraformaldehyde. If cells were grown on a feeder layer, feeders were removed prior to fixation. Cells were then stained using the same protocol as for tissues.

Micro-dissected pilosebaceous units were labelled using the same procedure as for tissue sections, except that hair follicles were suspended in microcentrifuge tubes instead of being placed on slides. Stained pilosebaceous units were mounted on slides in glycerol for imaging.

Primary antibodies were used at the indicated dilutions: GATA6 (1:200, D61E4 clone, Cell Signalling 5851); ITGα6 (1:200, GoH3 clone, eBioscience 14-0495-82); IVL (1:200, clone SY7 clone, CRUK); FASN (1:100, Santa Cruz sc-48357); KRT7 (1:100, LK1K clone, Thermo Fisher Scientific MA1-90894); KRT14 (1:200, Covance, SIG-3476-100); PanKer (1:400, clone LP34, LSBio LS-C95318); LOR (1:200, Covance, PRB145P); PLET1 (1:100, 1D4 clone, EMD Millipore MAB4416); P-SMAD2/3 (Phospho-SMAD2 (Ser465/467)/SMAD3 (Ser423/425)) (1:100, D27F4 clone, Cell Signalling 8828); P-SMAD2 (Phospho-SMAD2 (Ser465/Ser467)) (1:100, E8F3R clone, Cell Signalling 18338); KRT79 (1:100, Santa-Cruz, sc-243156); PPARγ (1:100, E-8 clone, Santa Cruz, sc-7273); LRIG1 (1:50, R&D Systems, MAB7498); SOX9 (1:100, R&D Systems, AF3075); BLIMP1 (1:100, 6DE clone, eBioscience 14-5963-82); AR (1:100, AN1-15 clone, Santa Cruz, sc-56824); KI67 (1:100, 8D5 clone, Cell Signalling 9449). All primary antibodies have been validated by the suppliers. Validation data can be found on the manufacturers' websites by searching for the antibody catalogue numbers. Alexa Fluor-conjugated secondary antibodies (ThermoFisher Scientific) were used at 1:1000 as secondary antibodies. Where indicated, dyes were incubated with secondary antibodies: HCS LipidTOX Deep Red Neutral lipid stain (1:500, Life Technologies) and Rhodamine Phalloidin (1:500, Life Technologies). TUNEL assay was performed using the In Situ Cell Death Detection Kit, TMR red (Roche) following the manufacturer's instructions. Images were acquired with a Nikon A1 confocal microscope or an Operetta high content imaging system (Perkin Elmer).

**Histopathological analyses**. Skin samples were fixed in 4% neutral buffered paraformaldehyde solution at room temperature for at least 24 h, embedded in paraffin blocks (Shandon Cytoblock, Thermo Scientific, USA), cut into 4 μm sections and stained with haematoxylin, eosin and saffron (H&E). Immunohistochemical procedures were carried out using an automated immunohistochemical apparatus according to the manufacturer's instructions (Bond-Max slide stainer, Leica Microsystems).

Briefly, after dewaxing, paraffin sections were rehydrated, and antigen retrieval was performed in an Antigen Retrieval Buffer (Leica Biosystems) at pH 9. Sections were incubated for 30 min with primary antibodies, rinsed, and then incubated with a biotinylated secondary antibody. Sections were rinsed and the reaction was developed according to the manufacturer's guidelines (streptavidin-peroxidase with an automated BOND, Leica Microsystems). Primary antibodies were used at the indicated dilutions: GATA6 (1:750, D61E4 clone, Cell Signalling 5851), Ki67 (1:100, Clone MIB-1, Dako M7240) and KRT5/6 (1:100, Clone D5/16 B4, Dako M7237). Biotinylated secondary antibodies (Vector Laboratories) were used at 1:400 dilution. Images were acquired with a Pannoramic 250 Flash slide scanner (3DHistech) and visualised with QuPath version 0.1.2 (https://qupath.github.io).

**Immunofluorescence microscopy of sebaceous organoids**. Sebaceous organoids were washed twice with ice-cold PBS. Matrigel droplets were lifted using a 1 ml syringe plunger (VWR), transferred to 0.5 ml ice-cold Cell Recovery Solution

(Corning) and incubated for 30 min on ice. Pellets were resuspended in ice-cold DMEM, then fixed in 4% paraformaldehyde for 45 min on ice. Organoids were then permeabilised and blocked for 1 h in freshly prepared PBO buffer (1% Triton X-100, 1% DMSO, 1% BSA and 5% donkey serum in PBS (all reagents from Sigma-Aldrich)).

Primary antibodies were diluted at 1:100 in PBO buffer and incubated overnight with organoids at 4 °C. Washes and incubation with secondary antibodies (diluted at 1:200) and DAPI were performed in PBS with 0.05% BSA for 2 h. Low-binding tips, tubes and plates were used throughout the protocol. Z-stacks were acquired with a Nikon Eclipse Ti inverted spinning disk confocal or a Leica TCS SP8 confocal microscope.

**Quantitative real-time RT-PCR.** Total RNA from cultured cells was isolated using the Qiagen RNeasy Mini Kit (Qiagen). Total RNA from sebaceous organoids was isolated by disaggregating the organoids in 1× TrypLE Express Enzyme (Gibco) for 20 min at 37 °C and processing them using the Qiagen RNeasy Micro Kit (Qiagen). cDNA was synthesised using the QuantiTect Reverse Transcription kit (Qiagen). Quantitative real-time reverse transcriptase polymerase chain reactions (qRT-PCR) were performed on a CFX384 Real-Time System (Bio-Rad Laboratories) using SYBR-Green Master Mix (Life Technologies) and qPCR primers designed with Primer3Plus version 2 (http://www.bioinformatics.nl/cgi-bin/primer3plus/primer3plus.cgi). Values were normalised to housekeeping gene (*18s*, *GAPDH*, *RPL13*, and/or *TBP*) expression levels using the CT method. For each biological replicate, the reaction was performed in technical duplicates. Primers are listed in Supplementary Data 3.

**Flow cytometry.** For cell cycle analysis, isolated sebocytes were fixed with ice-cold 70% EtOH for 30 min at 4 °C, washed twice with PBS and treated with 50 μl of 100 μg.ml$^{-1}$ RNAse A solution (Sigma-Aldrich) for 15 min at room temperature. Cells were then passed through a 70 μm strainer and incubated with 200 μl of 50 μg ml$^{-1}$ propidium iodide solution (Sigma-Aldrich) for 10 min at room temperature. Measurements were performed with a BD FACSCanto II cell analyser (gating strategy is shown in Supplementary Fig. 9a).

For Fig. 6c and Supplementary Fig. 7a, disaggregated cells were resuspended in FACS buffer (3% foetal bovine serum and 1 mM EDTA in PBS) and blocked with 5 μl Human TruStain FcX (Fc Receptor Blocking Solution) (BioLegend) for 20 min at 4 °C. The following primary antibodies were incubated with the cells for 20 min at 4 °C in FACS buffer: CD86—PE (1:50, IT2.2 clone, Biolegend, 305405); CD40—APC/Cy7 (1:50, 5C3 clone, Biolegend, 334323); HLA-DR—Pacific Blue (1:50, LN3 clone, Biolegend, 327016); CD80—Brilliant Violet 605 (1:50, 2D10 clone, Biolegend, 305225); CD274/PD-L1/B7-H1—PE-Cy7 (1:50, MIH1 clone, eBioscience, 25-5983-41). After incubation, cells were washed twice in FACS buffer and passed through a 70 μm strainer. For gate setting and compensation, unlabelled cells and single-labelled BD CompBeads (BD Biosciences) were used as controls. Gating strategy is shown in Supplementary Fig. 9b. Data acquisition was performed using a LSRFortessa cell analyzer.

For all flow cytometry experiments, data analysis was performed using FlowJo software version 10.4 (FlowJo).

**Cell fractionation and western blotting.** For nuclear/cytoplasmic fractionation, cells grown in a 10 cm-dish were washed twice with PBS on ice and scraped into Subcellular Buffer (250 mM Sucrose, 10 mM HEPES pH 7.4, 5 mM KCl, 1.5 mM MgCl$_2$, 1 mM EDTA). Lysates were passed ten times through a 26 Gauge needle, incubated for 15 min on ice, and centrifuged at 700×g for 5 min at 4 °C. The supernatant was kept as the cytosolic fraction, while the pellet was washed twice with buffer F1 (20 mM Tris pH 7.4, 0.1 mM EDTA, 2 mM MgCl$_2$) and three times with buffer F1 supplemented with 0.7% CHAPS. After spinning down at 600×g for 5 min, the pellet was resuspended in buffer A (20 mM HEPES, 0.4 M NaCl, 2.5% glycerol, 1 mM EDTA, 0.5 mM NaF) and kept as the nuclear fraction. All buffers were supplemented with Complete Protease Inhibitor Cocktail (Roche), PhosSTOP Phosphatase Inhibitor Cocktail (Roche) and 1 mM DTT. Unless specified, all reagents were purchased from Sigma-Aldrich.

Total protein amounts were quantified using the BCA kit (Pierce). 10 μg protein extract was resolved by SDS-PAGE in 4–20% Mini-PROTEAN® TGX™ Precast Protein Gels (Bio-Rad Laboratories) and transferred onto Trans-Blot 0.2 mm PVDF membranes (Bio-Rad Laboratories) using the Trans-Blot Turbo transfer system (Bio-Rad Laboratories). Precision Plus Protein Dual Colour Standards were used as molecular weight ladders (Bio-Rad Laboratories). Membranes were blocked with 5% non-fat milk (Tebu-Bio) in PBS supplemented with 0.05% Tween-20 (Sigma-Aldrich) and then probed with GATA6 antibody diluted at 1:1000 in blocking buffer. Primary antibody-probed blots were visualised with appropriate horseradish peroxidase-coupled secondary antibody (Jackson ImmunoResearch) using Clarity Western ECL enhanced chemiluminescence (Bio-Rad Laboratories) according to the manufacturer's instructions.

For the cytokine assay, cells were lysed in 1× RIPA buffer (Cell Signalling) supplemented with Complete Protease Inhibitor Cocktail and PhosSTOP Phosphatase Inhibitor Cocktail on a spinning wheel for 1 h at 4 °C. Supernatants were quantified using the BCA kit (Pierce) and 300 mg of total protein was applied to a Human Cytokine Array (R&D Systems) following manufacturer's instructions.

Protein bands and cytokine arrays were detected using a ChemiDoc Touch Imaging System (Bio-Rad Laboratories). Processing of western blot images was performed with Image Lab software (Bio-Rad Laboratories).

**Image analysis.** Digital images from fluorescence microscopy experiments were processed using NIS elements Advanced Research version 5.11.01 (Nikon), NIS elements Viewer version 4.11.0 (Nikon), ImageJ version 2.0.0-rc-69/1.52p (https://imagej.nih.gov/ij/) or the Harmony high content analysis version 4.1 software package (Perkin-Elmer). When performed, image processing was restricted to changes in brightness and/or contrast and was applied equally across the entire image and different images of the same experiment, including controls.

For image analysis, at least three fields per biological replicate were randomly chosen.

To measure positive cells, NIS elements Advanced Research was used. Maximal intensity projections were obtained. Spot detection was applied to detect bright spots with a typical diameter of 10 μm and above a certain intensity threshold for the DAPI channel. Object counts of DAPI spots gave the total number of cells. A similar process was applied to detect objects in the other channels (typical diameter between 10 and 20 μm depending on nuclear or cytoplasmic staining). Thresholds were determined per staining and kept similar between conditions within the same experiment. The percentage of positive cells was determined by the ratio of positive cells over the total number of cells.

To measure staining intensity, the Harmony high content analysis software package was used with custom algorithms. The DAPI channel was used to detect nuclei. Cytoplasmic area was determined as a ring of constant size around each nucleus and was used to measure the mean fluorescence intensity of the staining of interest.

For clonogenicity assays, colony detection was performed using a home-made Python script that automatically detects colonies from 6-well plates. Briefly, areas of interest were selected, a threshold was applied to detect colonies and noise was removed. Individual colonies were then detected (even from merged colonies), and their size and number were measured. Full script is available in GitHub (https://github.com/MATBEO/clonogenicity_assay_script).

For IncuCyte Zoom analysis, one phase-contrast image was acquired every hour per field (at least two fields were acquired per well). Confluence was measured as a percentage of the surface occupied by cells detected by a mask in the IncuCyte Zoom software version 2018A (Essen Bioscience).

For analysis of cytokine arrays, the intensity of each spot was measured using Image Lab software version 4.1 (Bio-Rad Laboratories). Background was removed from all values, and they were normalised to the positive control spots. Quantitative data are presented only for the targets reaching a level of expression above background for all conditions.

**Computational analysis.** We performed unsupervised clustering and differential gene expression analyses of the previously published single cell RNA sequencing dataset EGAS00001002927[29] in the Seurat R package v2.3.0[72] using R version 3.6.1 (https://www.r-project.org). We reanalysed the previously published microarrays GEO GSE53795[31] and GSE6475[30] using NCBI GEO2R tool (https://www.ncbi.nlm.nih.gov/geo/geo2r/). Upstream regulator analysis was performed using Ingenuity Pathway Analysis (IPA) software version 01-13 (Qiagen). KEGG pathway and Gene Ontology analyses were performed using DAVID software version 6.8 (https://david.ncifcrf.gov). Network visualisation was performed using String software versions 10.5 and 11 (https://string-db.org) and microarray gene expression overlay of signalling pathways using IPA and R.

**Statistics and reproducibility.** Statistical analyses were performed using Prism 7 and 8 software (GraphPad). No statistical method was used to predetermine sample size. The number of independent experiments and of replicates (*n*) is indicated in the figure legends. Unless stated otherwise, at least three biological replicates were performed for each panel and came from at least two independent experiments. When appropriate, normalisation of the data was performed within each independent experiment. When required, a statistical method to correct for multiple comparison was used as recommended by Prism software.

**Reporting summary.** Further information on research design is available in the Nature Research Reporting Summary linked to this article.

## Data availability

All data that support the findings of this study are available within the paper and its supplementary information files or are available from the corresponding author upon reasonable request. Publicly available datasets used in this study were: EGAS00001002927[29] (https://ega-archive.org/datasets/EGAD00010001620), GEO GSE53795[31] and GSE6475[30]. Source data are provided with this paper.

## Code availability

The home-made Python script developed to quantify clonogenicity assays is available in GitHub (https://github.com/MATBEO/clonogenicity_assay_script).

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

## Acknowledgements

We are grateful to all members of the Watt laboratory for helpful discussions. We thank Simon Broad and Ajay Mishra for constant help and advice; Rocio Sancho for helping with organoid protocols; Arsham Ghahramani for helping with computational analysis; and Mark Soldin for providing skin samples. We thank Anne Couvelard, Karim Tabbech and Amina El Hilali from the Pathology Department of Hôpital Bichat (AP-HP, Hôpitaux Universitaires Paris Nord Val de Seine, France) for histopathological analyses. We are grateful for funding from the Department of Health via the National Institute for Health Research comprehensive Biomedical Research Centre (BRC) award to Guy's & St Thomas' National Health Service Foundation Trust in partnership with King's College London and King's College Hospital NHS Foundation Trust. We are grateful to the BRC Flow Cytometry Facility, King's College Hospital/NHS Foundation Trust and to the Nikon Imaging Centre, King's College London, for technical support. This work was funded by grants to F.M.W. from the UK Medical Research Council (G1100073), the Wellcome Trust (096540/Z/11/Z) and from Unilever.

## Author contributions

B.O. and F.M.W. conceived the study and designed the experiments. B.O., C.P., M.T., M.V.R., A.M.C., L.D., and G.D. performed experiments and analysed data. J.S. and M.T. performed computational analysis. M.T. wrote the Python script for clonogenicity image analysis. P.A.G. and N.D. provided *P. acnes* cultures. S.Q., K.N. helped with histo-pathological analyses. B.O., C.P., and F.M.W. wrote the manuscript with inputs from all the authors.

## Competing interests

The authors declare no competing interests.
