## [Peer Review File · Nature Communications]

Reviewers' comments:

Reviewer #1 (Remarks to the Author):

Introduction:

-- What is currently known about the pathophysiology of acne vulgaris is synthesized too sketchily and insufficiently comprehensively (major, well-supported pathobiology scenarios are ignored). From a study that claims to shed important new light on this, one expects a more compelling synthesis of the current state-of-the-art of acne pathobiology.

-- The specific questions addressed in this study are not clearly defined, and the rationale of the experimental design chosen to answer these questions remains quite unclear.

-- As the authors acknowledge themselves, acne does not occur in mice (which questions the use of mice in general as an instructive model system for studying acne), while a cultured human sebocyte cell line cannot possibly reflect the complex pathobiology of acne, with multiple different cell populations and tissues interacting in a 3D context. Therefore, it is particularly important to provide a convincing rationale for the chosen study design, whose immediate relevance for acne research is at the very least debatable.

Results:

-- For which cells/structures exactly in the upper human PSU is GATA6 a (specific?) marker, and how has this been unequivocally demonstrated?

-- The authors present no evidence that the downregulation of GATA6 in (very few!) examined acne samples is more than an epiphenomenon, resulting e.g. from the strong proinflammatory milieu associated with acne lesions, rather functionally important for acne development.

-- At the very least one would need to see the dynamics of GATA6 downregulation in very acute compared to chronic acne lesions as well as in non-lesional compared to healthy human skin to generate moderately suggestive evidence in human skin samples (rather than mice or cultured cells) that GATA6 downregulation plays any important role in the critical early stages of acne pathobiology.

-- Direct evidence that GATA-low ORS keratinocytes in acne hair follicles are hyperproliferative or show keratinisation/differentiation abnormalities in situ is missing (or did I overlook something?).

-- Instead of using upper ORS keratinocytes to explore the role of GATA6 in human hair follicle keratinocyte proliferation, epidermal keratinocytes are employed, even though these cells are biologically quite different from upper hair follicle keratinocytes (see Minor Comments) and not known to play a role in acne pathogenesis.

-- GATA6 effects on the terminal differentiation of human upper ORS keratinocytes, which would have been important to study given the hyperkeratinization seen in early acne lesions, are not studied.

-- Given that gene knockdown is possible in organ-cultured human hair follicles and in full-thickness human skin containing SGs (published repeatedly), the only convincing evidence for the acne-related pathogenesis theory proposed by the authors would arise from silencing GATA6 in a) microdissected full-length human hair follicles and b) SG-rich human skin ex vivo, utilizing well-established protocols.

-- While suggestive, I do not quite see how any of the presented SebE6E7 data (i.e. in a transformed cell line cultured under highly artificial conditions) confirms the claim that "Loss of GATA6 contributes to acne pathogenesis in human skin" and that "GATA6 is a master regulator of JZ/SG homeostasis in human skin". Rather, these data show that GATA6 plays multiple regulatory role in the biology of this cell line and its response to retinoic acid, P. acnes, and IFN γ in vitro (some of which may arguably be relevant to acne; yet without more direct human skin, hair follicle and SG evidence for this, the above claims remain very tenuous).

-- As the authors acknowledge themselves, acne pathogenesis likely begins in the upper ORS of the hair follicle and may involve inappropriate differentiation of SG progenitor cells within or close to this zone. Therefore, while biologically interesting as such, the 2D and 3D culture of a sebocytes (namely of a cell line) cannot possibly shed light on what happens in this part of the hair follicle epithelium in previously healthy skin during puberty, giving rise to acne lesions such as comedones and folliculitis.

-- Again, human skin and hair follicle organ culture, and/or - even better - the use of human skin xenotransplants onto immunocompromised mice would have been the appropriate model systems. Certainly, evidence from these models would be required to support the bold claim that the current "work identifies GATA6 as an essential regulator of the human upper pilosebaceous unit in humans [sic]"

Discussion:

-- Even if one accepts the above claim for argument's sake, after reading the Discussion, I remain quite confused as to what exactly the molecular and cellular scenario is that the authors hypothesize, on the basis of the presented data, to lead to infundibular hyperkeratinization and/or hyperproliferation and ultimately comedo formation and a neutrophilic folliculitis, the key characteristics of acne. A carefully designed, sufficiently detailed cartoon that synthesizes the postulated choreography of pathobiology events is needed here to help the reader understand the pathobiology scenario that is proposed.

Minor comments:

-- A bit more detail on SG anatomy/histology & biology and the differences between murine and human SGs (Introduction) would really be helpful.

-- It is a frequently reverberated myth that the so-called "permanent" part of the hair follicle "does not undergo growth or regression during the HF cycle". It actually shows discrete, but significant hair cycle-dependent changes in apoptosis, at least in mice (Lindner et al. Am J Pathol 1997).

-- The Watt Lab frequently calls the epithelial component of the hair follicle "epidermal". This misleading choice of terminology has caused much confusion, since it breeds the misunderstanding that epidermal and hair follicle keratinocytes are very similar - which, in fact, they are not (besides major differences in keratin expression patterns and the microbiota they harbor, the former do not normally give rise to sebocytes and are independent of hr signaling, while the latter totally depend on hr activity and harbor progenitor cells for the SG; also, there are major differences in the neuroendocrinology, immunology, chemokine, and antimicrobial peptide secretion of these two very distinct keratinocyte populations, and in their response to neurotrophins, to name but a few selected examples).

Reviewer #2 (Remarks to the Author):

The results presented here provide valuable insight into acne pathogenesis for the scientific community. Despite the large number of people impacted by acne, the mechanisms of acne induction remain unclear. Thus, any study describing novel mechanisms or potential therapeutic targets will be of interest to a sizable lay audience in addition to cutaneous/stem cell biologists and dermatologists. Overall, the manuscript is well written, the data are solid and convincing, and the findings are exciting. The experiments are nicely constructed and thorough; therefore, I primarily have minor comments and suggestions worth addressing prior to publication:

Specific comments:

1. Evidence for GATA6 down-regulation in patient samples: Overall, the data in Figure 1 are quite convincing, however, one worry would be that the acne tissue samples (Fig. 1F/Right) appear to lack GATA6 expression due to a processing error (e.g. over-fixation). Have the authors ensured that lack of GATA6 protein expression is not due to fixation issues? To remove all doubt, the authors could include lower magnification images of the comedones. Do "healthy" follicles neighboring comedones display a normal GATA6 expression pattern in these samples?

2. Colony analysis in Fig 2d-f: Was the starting number of colonies the same in every condition being compared? Would the colony number in GATA6 + vehicle and Mock + vehicle be the same? There seems to be a large difference in colony numbers between those two groups. It shouldn't be an effect of GATA6 as the cells were not exposed to DOX; is this an effect of contamination? Should the

vehicle vs Dox in Mock and vehicle vs Dox in GATA6 be compared, as well? Please include a more thorough explanation of the experimental and analysis procedures for these data.

3. shGATA6 experiments: Have the authors attempted a rescue experiment of GATA6 in the shGATA6 cell line to determine whether shGATA6 cells can recover to normal with externally introduced GATA6?

4. Immune response data (Fig. 5): This was a weak point of the manuscript. I commend the authors on the depth of analysis; however, the effects of GATA6 and the various treatments appear quite subtle and thus difficult to interpret. The authors should revisit this section to better incorporate the findings into the narrative. Have the authors tested a GATA6-knockdown cell line to compare the immune system response/regulation and confirm the involvement of GATA6 in this response?

5. Organoid experiments: The 3D assay in figure 6 is quite convincing. The authors may want to confirm results from earlier 2D culture experiments (Fig. 2-4) using this assay to strengthen their findings. One potential issue with the organoid data is that there appears to be a discrepancy between RepSox treated organoid size between Figure 6f and 6g. Are all SG organoid sizes within a group consistent? Please clarify. It may be worthwhile to quantify organoid size re: Figure 6b.

6. Page 12, pSMAD analysis: Why did the authors go back to 2D? What would it be like in 3D, which better recapitulates native SGs?

Readability Suggestions:

1. To help the non-expert reader, add a Figure 1a and b, prior to presenting your IHC images, with schematic images of: a) a pilosebaceous unit, annotating each compartment (e.g. cartoon with labeling of INF, IFE, JX, SG, HF/SH, and clear demarcation of the 'upper' pilosebaceous unit), and b) sebaceous gland development and structure (e.g. proliferative cells and lipid-rich sebocytes, upper or lower SG, sebaceous ducts, at early and late stages, ending with sebum release)

2. Again, in Figure 6 (or a last Figure), it would be helpful to see an overall summary schematic image of mechanism/signaling pathway that describes loss of GATA6 leading to acne (e.g. low vs high GATA6 -> TGFbeta -> pSMAD2/3 -> acne vs normal + immune system regulation; i.e. every signaling/mechanism involved in acne pathogenesis that is investigated in the manuscript)

3. More white space in Figures 2-6 would be helpful.

4. For all IHC data, the overlap of Cyan (nuclear protein) and White (DAPI) is very difficult to interpret. Consider a different color combination.

5. Double check labels for consistency between text and figures: e.g. page 12 "CK7" and Fig 6g "KRT7"

6. Page 6, 2nd line: "Furthermore, GATA6 expression was drastically reduced..." In which condition/anatomical location?

7. Page 6, 6-9 lines: "...GATA6 expression was decreased [IN PATIENTS?], although the difference was only significant in the larger dataset".
8. Double check all the figure numbers referenced in the text (e.g. page 6, last sentence: Supplementary Fig 1j. doesn't seem correct).
9. Supplementary Figure 2b and c: label concentration of Dox
10. Page 7, start of the section "GATA6 mediates the junctional zone and xxx": Add a first sentence describing the reason for testing the treatment of TRO w/wo LG.
11. Supplementary Figure 3: Add a high magnification image of TRO+LG next to the existing TRO+LG image to provide the reader with a clearer impression of a differentiated SebE6E7 cell with lipid vacuoles.
12. Fig. 3a and d: Change colors; IVL and LOR (also, KRT7 and PLET1) both are cytosolic staining that if any cells express both, it cannot be clarified and not convincing with current colors. Consider changing to red and green.
13. Fig 3d: The intensity of KRT7 and PLET1 both look dramatically higher in the GATA6 overexpressed condition. Any explanation or analysis of intensity comparison available?
14. Make sure to label all treatments in detail for clear understanding: e.g. Fig. 3f and Fig. 4e; label as "+Dox +Vehicle" & "+Dox +TRO+LG" and "+Dox +Vehicle" & "+Dox +RA 5uM".
15. Page 11, end of first paragraph, "our results suggest a new...": This is a vague sentence, please inject more detail (e.g. ...upper pilosebaceous unit "via activation/suppression of IFN γ /HLA-DR/CD40").
16. Page 12, 2nd paragraph, "Immunostaining of sebaceous organoids for PLET1 revealed similarities between...": The comparison is not clear in this sentence—exactly which groups are you comparing (shGATA6 expressing vs. shGATA6+RepSox? Or control+RepSox? Or all of RepSox treated groups?)—either way, they look different. shGATA6 size is smaller than all RepSox treated groups). In the following sentence, "controls" and "they" do not clearly indicate which groups is being referenced (comparison of each RepSox treated groups to each vehicle treated groups? Or all RepSox treated groups to vehicle treated control group?). Please clarify.
17. Fig 6f & g: Re-label the treatments (e.g. +Dox +Vehicle, +Dox +RepSox).

Reviewer #3 (Remarks to the Author):

This is a very good piece of work, which suggests that GATA6 is a key regulator of homeostasis in the upper part of the pilosebaceous unit using several in tissue and in vitro techniques including functional studies. The authors provide strong evidence for their conclusions. The most part of the work is novel and of extreme importance to scientists in the specific field. Moreover, it is interesting to researchers in other related disciplines. On the other side, the authors claim that GATA6 may be involved in the development of acne lesions and an actionable target in the treatment of acne. Despite some well worked confirming data, Swanson JB et al have already shown that loss of Gata6 causes dilation of the hair follicle canal and sebaceous duct (Swanson JB et al. Loss of Gata6 causes dilation of the hair follicle canal and sebaceous duct. *Exp Dermatol* 28:345-349, 2019). Such findings can be seen in seborrheic skin but not obligatorily in acne. Similar findings are also described in exogenously induced acne (e.g. dioxin-induced acne) and in syndromes of aged, sun exposed skin (e.g. Favre-Racouchot disease). In contrast, acne is also associated with reduction of the free hair follicle canal due to hyperkeratosis and not only with a secondarily occurring dilation of the hair follicle canal. Moreover, GATA6 has been shown in previous work of the same authors and others (Oulès B et al. Mutant Lef1 controls Gata6 in sebaceous gland development and cancer. *EMBO J* 38(9), pii: e100526, 2019 / Siltanen S et al. Transcription factor GATA-6 is expressed in malignant endoderm of pediatric yolk sac tumors and in teratomas. *Pediatr Res* 54(4):542-546, 2003) to be involved in tumorigenesis. How acne and skin tumour induction by the same gene can be associated, when no tumour comorbidity exists in acne?

Page 3, line 19ff: The mentioning of four pathogenetic factors of acne represents an old not any more valid theory of acne pathogenesis. The authors should refer to the current knowledge of acne pathogenesis (Zouboulis CC, Dessinioti C. A new concept of acne pathogenesis. In: Zouboulis CC et al (eds) *Pathogenesis and Treatment of Acne and Rosacea*. Springer, Berlin Heidelberg New York, pp 105-107, 2014).

Page 3, line 23; page 9, line 24; page 10, lines 12, 15, 16; page 14, line 17: "...Cutibacterium acnes (C. acnes) species (previously Propionibacterium acnes)." There is no obligatory, clinically relevant change of the terminology of Propionibacteria and, therefore, there is no need of substitution of the term "Propionibacterium acnes" by any other term (Alexeyev OA et al. Why we continue to use the name Propionibacterium acnes. *Br J Dermatol* 179:1227, 2018).

Page 4, lines 4-5: "Nevertheless, recent studies in mice have provided insights into the development and function of the pilosebaceous unit." The development of the pilosebaceous unit is also described in humans long ago. Morphogenesis and human sebaceous differentiation have been repeatedly reported (e.g. Zouboulis CC et al. Sebaceous Glands. In: Hoath SB, Maibach HI (eds) *Neonatal Skin – Structure and Function*. 2nd ed, Marcel Dekker, New York Basel, pp 59-88, 2003; Liakou AI et al. Marked reduction of number and individual volume of sebaceous glands in psoriatic lesions. *Dermatology* 232:415-424, 2016). The mouse-originating information on molecular pathways may provide additional functional information at the molecular level, in case that the mouse data would be shown in the future to be representative for human functions.

Page 4, line 12: Not retinoids but specifically isotretinoin is the most effective treatment for acne vulgaris and its effectiveness is not currently detected but since 40 years.

Page 4. Line 13: The ref. #22 has to be substituted by a more specific one, e.g. Zouboulis CC. Isotretinoin revisited – Pluripotent effects on human sebaceous gland cells. *J Invest Dermatol*

126:2154-2156, 2006 / Zouboulis CC. The truth behind this undeniable efficacy – Recurrence rates and relapse risk factors of acne treatment with oral isotretinoin. *Dermatology* 212:99-100, 2006 / Ganceviciene R, Zouboulis CC. Isotretinoin: state of the art treatment for acne vulgaris. *Expert Rev Dermatol* 2:693-706, 2007.

Page 4, lines 17-18: “Since GATA6 plays a role in homeostasis of the upper HF in mouse, in the present report we have explored the potential link between GATA6 and acne pathogenesis.” The consequence cannot be considered as straight ahead, since the mouse does not develop acne even if GATA6 may play a role in homeostasis of its upper HF. None of the clinical characteristics of mouse HF changes resembles follicular inflammation in humans. Therefore, the authors should present another argumentation for their decision to study GATA6 expression changes and acne.

Page 5, line 23: Please use the term “Ki67” instead of “KI67”.

Page 6, line 3: Were the comedone-bearing patients not patients with acne vulgaris? Acne comedonica is a type of acne vulgaris, however, several disorders with comedones exist, which are acneiform diseases but not acne vulgaris. What was the correct diagnoses of the 4 patients?

Page 7, section “GATA6 mediates the effects of retinoic acid on sebocytes”: It is a pity that the authors only have used all-trans-RA in their experiments. There is, indeed evidence that 13-cis-RA (isotretinoin) has to be intracellularly metabolized to all-trans-RA to become active on human sebocytes (Tsukada M et al. 13-cis Retinoic acid exerts its specific activity on human sebocytes through selective intracellular isomerization to all-trans retinoic acid and binding to retinoid acid receptors. *J Invest Dermatol* 115:321-327, 2000) but since isotretinoin is the most potent systemic drug for acne, the detection of isotretinoin effects similar to those detected by all-trans-RA would better corroborate the conclusion of the authors.

Page 9, section “Immune regulation of the upper pilosebaceous unit is controlled by GATA6”: In all previous articles using human sebocytes as targets or partners of inflammatory response arachidonic acid has been used as trigger factor and not PGN. Why the authors made the PGN experiments, which cannot be compared with any previous studies?

Page 9, line 24; page 14, line 17: “To mimic *C. acnes* infection, ...”: Acne is not an infectious disease and *P. acnes* does not induce an infection in acne (Xu DT et al. Is human sebum the source of skin follicular ultraviolet-induced red fluorescence? A cellular to histological study. *Dermatology* 234:43-50, 2018).

Page 11, lines 8-9; page 12, lines 23-24: “For this reason, we developed a 3D sebaceous organoid model to better recapitulate SG homeostasis” and “By developing a new sebaceous organoid model, ...”: The described organoids have been previously published in Yoshida GJ et al. Three-dimensional culture of sebaceous gland cells revealing the role of prostaglandin E2-induced activation of canonical Wnt signaling. *Biochem Biophys Res Commun* 438:640–646, 2013.

Page 13, lines 4-6: “...but limits, in parallel, cell proliferation and lipid production to prevent hyperseborrhoea. It also orients sebocytes towards an anti-inflammatory phenotype.”: If GATA6 can, indeed, reduce lipogenesis and induce an anti-inflammatory phenotype in acne, it should also modify the lipid fractions produced by the human sebocytes from pro-inflammatory to anti-inflammatory ones (Zouboulis CC et al. Acne is an inflammatory disease and alterations of sebum

composition initiate acne lesions. *J Eur Acad Dermatol Venereol* 28:527-532, 2014 / Töröcsik D et al. Leptin promotes a pro-inflammatory lipid profile and induces inflammatory pathways in human SZ95 sebocytes. *Br J Dermatol* 171:1326-1335, 2014 / Lovászi M et al. Sebaceous immunobiology is orchestrated by sebum lipids. *Dermatoendocrinol* 9:e1375636, 2017). Such experiments should be additionally performed to corroborate this conclusion.

Page 13, line 9: "Acne mostly occurs in adolescents" reads more correctly than "Acne is a disease that occurs in teenagers when puberty is initiated."

Page 13, lines 11-12: "...the excellent therapeutic effect of combined oral contraceptive pills." Acne only responds to hormonal antiandrogens and not to every contraceptive pill (Dessinioti C, Zouboulis CC. Hormonal therapy for acne. In: Zouboulis CC et al (eds) *Pathogenesis and Treatment of Acne and Rosacea*. Springer, Berlin Heidelberg New York, pp 477-482, 2014).

Page 13, lines 12-14: "...patients with acne frequently have clinical signs of hyperandrogenism and increased serum and skin levels of testosterone and associated androgens such as dehydroepiandrosterone." This is not true. Patients with acne have in maximum 20% clinical signs of hyperandrogenism but rarely hyperandrogenemia (Chen W, Zouboulis CC. Acne and androgens. In: Zouboulis CC et al (eds) *Pathogenesis and Treatment of Acne and Rosacea*. Springer, Berlin Heidelberg New York, pp 131-134, 2014).

Page 14, lines 23-24: "This suggests that immune overactivation is detrimental in acne and that anti-inflammatory treatments are needed.": Indeed, anti-inflammatory but not antibacterial (Zouboulis CC. Is acne vulgaris a genuine inflammatory disease? *Dermatology* 203:277-279, 2001 / Zouboulis CC. Zileuton, a new efficient and safe systemic anti-acne drug. *Dermatoendocrinol* 1:188-192, 2009).

Page 15, line 3-5: "In conclusion, our work identifies GATA6 as a crucial regulator of the upper pilosebaceous unit in human skin. Its loss is likely to contribute to the pathogenic features of acne, opening up new avenues for acne research and treatment." Indeed, this work clearly shows a major role of GATA6 in the proper anatomy and function of the acroinfundibulum. It also indicates that GATA6 may be involved in acne. However, previous work of the same authors and others (Oulès B et al. Mutant Lef1 controls Gata6 in sebaceous gland development and cancer. *EMBO J* 38(9), pii: e100526, 2019 / Siltanen S et al. Transcription factor GATA-6 is expressed in malignant endoderm of pediatric yolk sac tumors and in teratomas. *Pediatr Res* 54(4):542-546, 2003) indicate an association of GATA6 downregulation with tumorigenesis. How acne and skin tumour induction by the same gene can be associated, when no tumour comorbidity exists in acne?

Reference #4: The correct designation of the reference is: Moradi-Tuchayi, S. M. et al. Acne vulgaris. *Nat. Rev. Dis. Prim.* 1, 15029 (2015).

Reviewer #1:

Introduction:

-- What is currently known about the pathophysiology of acne vulgaris is synthesized too sketchily and insufficiently comprehensively (major, well-supported pathobiology scenarios are ignored). From a study that claims to shed important new light on this, one expects a more compelling synthesis of the current state-of-the-art of acne pathobiology.

The introduction has now been revised to include a more comprehensive synthesis of the literature on acne and models of its pathobiology.

-- The specific questions addressed in this study are not clearly defined, and the rationale of the experimental design chosen to answer these questions remains quite unclear.

We apologize for this lack of clarity. We have now explained more clearly the specific questions addressed in this study at the end of the Introduction. We have also better detailed our choice of experimental design.

-- As the authors acknowledge themselves, acne does not occur in mice (which questions the use of mice in general as an instructive model system for studying acne), while a cultured human sebocyte cell line cannot possibly reflect the complex pathobiology of acne, with multiple different cell populations and tissues interacting in a 3D context. Therefore, it is particularly important to provide a convincing rationale for the chosen study design, whose immediate relevance for acne research is at the very least debatable.

We agree with that acne is a disease specific to humans and that no animal model recapitulates its pathological features. Therefore, acne research relies on translational clinical studies and cell-based studies using primary sebocytes and sebocyte cell lines, such as SZ95. These approaches have provided valuable knowledge about sebaceous gland physiology and pathology. Given this, our study design uses human samples, 2D and 3D cellular models and, in the revised version, *ex vivo* organ-cultured hair follicle explants. This is now explained in the Introduction.

Results:

-- For which cells/structures exactly in the upper human PSU is GATA6 a (specific?) marker, and how has this been unequivocally demonstrated?

We thank the reviewer for this question. In the previous version of the manuscript, we only used immunostaining of healthy human skin to show that GATA6 is a marker of a cell population in the upper pilosebaceous unit. We have now added new analyses based on a single-cell RNA atlas of epithelial cells originating from three healthy human scalp samples published by the group of Raymond Cho (Cheng et al Cell Reports 2018). We performed an unbiased analysis of the scalp data and identified 24 clusters (new Fig. 1f). GATA6 was significantly and exclusively enriched in cluster 22 (new Fig. 1g), in which several genes of the INF/JZ (CST6, KRT79, ATP6V1C2) and of the SG (RBP1, SOX9, SQLE, DGAT2) were also significantly upregulated (new Fig. 1h). In addition, cluster 22 was relatively similar to the cluster "UHF Diff" of the differentiated upper hair follicle cells identified by Cheng et al (new Fig. 1i). We believe these new data, combined with the immunostaining, strongly support the conclusion that GATA6 is a marker of differentiating cells of the lower INF/JZ/SD and of the upper SG.

-- The authors present no evidence that the downregulation of GATA6 in (very few!) examined acne samples is more than an epiphenomenon, resulting e.g. from the strong

proinflammatory milieu associated with acne lesions, rather functionally important for acne development.

Again, we thank the reviewer for pointing out this weak point of the study. We have now collected new sections and analysed 8 healthy and 10 acne vulgaris patients presenting with differences in disease severity (New Fig. 2a and Supplementary Fig. 2). This confirmed that GATA6 was downregulated in early lesions (comedones) when the immune infiltrate (as evidenced by H&E staining) is not yet present or remains low (as in the first acne samples displayed in Fig. 2a and Supplementary Fig. 2).

-- At the very least one would need to see the dynamics of GATA6 downregulation in very acute compared to chronic acne lesions as well as in non-lesional compared to healthy human skin to generate moderately suggestive evidence in human skin samples (rather than mice or cultured cells) that GATA6 downregulation plays any important role in the critical early stages of acne pathobiology.

As explained above, we have now added new acne vulgaris staining showing lesions at different disease severity stages (New Fig. 2a and Supplementary Fig. 2). GATA6 expression was reduced in comedone walls of early lesions, and was completely lost in advanced lesions. It was difficult to find biopsies with non-lesional hair follicles as small diameter biopsies punches (3 mm) are usually used to perform biopsies on the face. However, we found that GATA6 expression was maintained in uninvolved hair follicles in the first acne section of new Fig. 2a.

-- Direct evidence that GATA-low ORS keratinocytes in acne hair follicles are hyperproliferative or show keratinisation/differentiation abnormalities in situ is missing (or did I overlook something?).

In addition to GATA6 labelling, we have stained healthy and acne vulgaris sections with anti-Ki67 as a marker of proliferation and anti-KRT5/6 for keratinisation (New Fig. 2a and Supplementary Fig. 2). As compared to the normal upper hair follicle walls, we observed that comedone walls were most often thicker (suggesting an increased proliferation) and more differentiated as evidenced by KRT5/6+ labelling. We also noticed an increase in Ki67+ cells in acne biopsies (New Fig. 2a and Supplementary Fig. 2).

-- Instead of using upper ORS keratinocytes to explore the role of GATA6 in human hair follicle keratinocyte proliferation, epidermal keratinocytes are employed, even though these cells are biologically quite different from upper hair follicle keratinocytes (see Minor Comments) and not known to play a role in acne pathogenesis.

We have now repeated the experiments using ORS keratinocytes (New Fig. 3c-d) and obtained similar results to those obtained with IFE keratinocytes. The new results thus corroborate the finding that overexpression of GATA6 reduces the proliferation of keratinocytes, including follicular keratinocytes.

-- GATA6 effects on the terminal differentiation of human upper ORS keratinocytes, which would have been important to study given the hyperkeratinization seen in early acne lesions, are not studied.

Our new colony forming assays show that GATA6 expression in ORS keratinocytes led to smaller size colonies, indicating not only a defect in proliferation but also in terminal differentiation (New Fig. 3d).

-- Given that gene knockdown is possible in organ-cultured human hair follicles and in full-thickness human skin containing SGs (published repeatedly), the only convincing evidence

for the acne-related pathogenesis theory proposed by the authors would arise from silencing GATA6 in a) microdissected full-length human hair follicles and b) SG-rich human skin *ex vivo*, utilizing well-established protocols.

We have now performed siRNA-mediated GATA6 knockdown in organ-cultured whole human hair follicle explants as originally pioneered and developed by the Paus lab (Langan et al *Exp Dermatol* 2015). We were unable to obtain SG-rich samples to perform the experiment suggested in (b).

GATA6 was efficiently knocked-down in microdissected full-length human hair follicles using siRNAs (new Fig. 4b and Supplementary Fig. 5b). Inhibition of GATA6 tended to increase IVL+ cells in the upper infundibulum (new Fig. 4b), however the effect was modest. As discussed in the manuscript, we believe this could be due to technical limitations of the model, namely the kinetics of gene knockdown and the time of culture required for hyperkeratinisation to occur.

-- While suggestive, I do not quite see how any of the presented SebE6E7 data (i.e. in a transformed cell line cultured under highly artificial conditions) confirms the claim that "Loss of GATA6 contributes to acne pathogenesis in human skin" and that "GATA6 is a master regulator of JZ/SG homeostasis in human skin". Rather, these data show that GATA6 plays multiple regulatory role in the biology of this cell line and its response to retinoic acid, P. acnes, and IFN γ *in vitro* (some of which may arguably be relevant to acne; yet without more direct human skin, hair follicle and SG evidence for this, the above claims remain very tenuous).

As discussed above, acne research is limited by the lack of an appropriate animal model or a recognized *ex vivo* model. However, the SZ95 sebocyte cell line has yielded significant insights into both sebaceous physiology and acne pathogenesis (Sneider & Zouboulis *Exp Dermatol* 2018). The SebE6E7 line is very similar to SZ95 (see, Lo Celso et al *Stem Cells* 2008, PMID: 18308950). In our manuscript, it is the combination of human samples, 2D and 3D cellular models using primary ORS or IFE keratinocytes, SebE6E7 sebocytes and, now, *ex vivo* organ-cultured hair follicle explants that allows us to make the claim that "Loss of GATA6 contributes to acne pathogenesis in human skin".

-- As the authors acknowledge themselves, acne pathogenesis likely begins in the upper ORS of the hair follicle and may involve inappropriate differentiation of SG progenitor cells within or close to this zone. Therefore, while biologically interesting as such, the 2D and 3D culture of a sebocytes (namely of a cell line) cannot possibly shed light on what happens in this part of the hair follicle epithelium in previously healthy skin during puberty, giving rise to acne lesions such as comedones and folliculitis.

We specifically chose to work with the immortalised SebE6E7 cell line as these cells differentiate into lipid-filled sebocytes or IVL+ follicular keratinocytes (Le Celso et al *Stem Cells* 2008). We postulate that these cells are derived from JZ/SD progenitors that can contribute both to the follicular and sebaceous compartments. We have now explained this more clearly in the text.

-- Again, human skin and hair follicle organ culture, and/or - even better - the use of human skin xenotransplants onto immunocompromised mice would have been the appropriate model systems. Certainly, evidence from these models would be required to support the bold claim that the current "work identifies GATA6 as an essential regulator of the human upper pilosebaceous unit in humans [sic]"

We believe that the new scRNAseq data (new Fig. 1f-j), histopathological studies (new Fig. 2a and Supplementary Fig. 2) and organ-cultured hair follicle explants (new Fig. 4b and Supplementary Fig. 5) better support our claim.

With regard to the use of acne sample xenotransplants onto immunocompromised mice, this is not widely used in acne research, to our knowledge, apart from in a study by Petersen et al (J Clin Invest 1984). We agree that this model could be very helpful, however maintaining prolonged GATA6 inhibition (as skin xenotransplants have to be kept on mice for about 2 months) would be technically difficult.

Discussion:

-- Even if one accepts the above claim for argument's sake, after reading the Discussion, I remain quite confused as to what exactly the molecular and cellular scenario is that the authors hypothesize, on the basis of the presented data, to lead to infundibular hyperkeratinization and/or hyperproliferation and ultimately comedo formation and a neutrophilic folliculitis, the key characteristics of acne. A carefully designed, sufficiently detailed cartoon that synthesizes the postulated choreography of pathobiology events is needed here to help the reader understand the pathobiology scenario that is proposed.

A cartoon of the proposed mechanisms has now been included (Fig. 8).

Minor comments:

-- A bit more detail on SG anatomy/histology & biology and the differences between murine and human SGs (Introduction) would really be helpful.

We have now added a schematic of SG anatomy/histology (new Fig. 1a) and new information about SG biology and the differences between murine and human SG in the Introduction.

-- It is a frequently reverberated myth that the so-called "permanent" part of the hair follicle "does not undergo growth or regression during the HF cycle". It actually shows discrete, but significant hair cycle-dependent changes in apoptosis, at least in mice (Lindner et al. Am J Pathol 1997).

This statement has now been corrected.

-- The Watt Lab frequently calls the epithelial component of the hair follicle "epidermal". This misleading choice of terminology has caused much confusion, since it breeds the misunderstanding that epidermal and hair follicle keratinocytes are very similar - which, in fact, they are not (besides major differences in keratin expression patterns and the microbiota they harbor, the former do not normally give rise to sebocytes and are independent of hr signaling, while the latter totally depend on hr activity and harbor progenitor cells for the SG; also, there are major differences in the neuroendocrinology, immunology, chemokine, and antimicrobial peptide secretion of these two very distinct keratinocyte populations, and in their response to neurotrophins, to name but a few selected examples).

The use of "epidermal" has now been corrected and replaced by "follicular", "ductal" or "epithelial", depending on the context.

Reviewer #2:

The results presented here provide valuable insight into acne pathogenesis for the scientific community. Despite the large number of people impacted by acne, the mechanisms of acne induction remain unclear. Thus, any study describing novel mechanisms or potential therapeutic targets will be of interest to a sizable lay audience in addition to cutaneous/stem

cell biologists and dermatologists. Overall, the manuscript is well written, the data are solid and convincing, and the findings are exciting. The experiments are nicely constructed and thorough; therefore, I primarily have minor comments and suggestions worth addressing prior to publication:

Specific comments:

1. Evidence for GATA6 down-regulation in patient samples: Overall, the data in Figure 1 are quite convincing, however, one worry would be that the acne tissue samples (Fig. 1F/Right) appear to lack GATA6 expression due to a processing error (e.g. over-fixation). Have the authors ensured that lack of GATA6 protein expression is not due to fixation issues? To remove all doubt, the authors could include lower magnification images of the comedones. Do “healthy” follicles neighboring comedones display a normal GATA6 expression pattern in these samples?

Further examination of new samples from acne patients and healthy controls has been carried out (new Fig. 2a and Supplementary Fig. 2). All samples were processed and stained at the same time to minimise technical errors and to allow direct comparisons. As stated above, it was difficult to find biopsies with non-lesional hair follicles as small diameter biopsies punches (3 mm) are used to perform biopsies on the face. However, we found that GATA6 expression was maintained in uninvolved hair follicles in the first acne section of new Fig. 2a.

2. Colony analysis in Fig 2d-f: Was the starting number of colonies the same in every condition being compared? Would the colony number in GATA6 + vehicle and Mock + vehicle be the same? There seems to be a large difference in colony numbers between those two groups. It shouldn't be an effect of GATA6 as the cells were not exposed to DOX; is this an effect of contamination? Should the vehicle vs Dox in Mock and vehicle vs Dox in GATA6 be compared, as well? Please include a more thorough explanation of the experimental and analysis procedures for these data.

The colony formation assay was repeated using keratinocytes isolated from the ORS (new Fig. 3d). For both IFE and ORS keratinocytes, equal numbers of cells were seeded in all conditions. However, slight differences in the number of attached cells (i.e. between Mock- and GATA6-infected cells, or between independent experiments) can occur given the very low number of seeded cells. For this reason, we normalised the number of colonies in Dox conditions over the corresponding vehicle conditions. There were no significant differences between Mock +/- Dox and GATA6 + vehicle conditions.

3. shGATA6 experiments: Have the authors attempted a rescue experiment of GATA6 in the shGATA6 cell line to determine whether shGATA6 cells can recover to normal with externally introduced GATA6?

Instead of doubly infecting cells with GATA6 and shGATA6 (which would be technically challenging and deleterious for cell viability), we treated cells with Retinoic Acid to mimic the addition of GATA6. We showed that GATA6 was induced in SebE6E7 cells upon RA treatment (Fig. 5b).

4. Immune response data (Fig. 5): This was a weak point of the manuscript. I commend the authors on the depth of analysis; however, the effects of GATA6 and the various treatments appear quite subtle and thus difficult to interpret. The authors should revisit section to better incorporate the findings into the narrative. Have the authors tested a GATA6-knockdown cell line to compare the immune system response/regulation and confirm the involvement of GATA6 this in this response?

This section of the manuscript has been entirely revised with the aim of clarifying our findings. New data analysing the expression of cytokines and AMP in shGATA6-sebocytes have also been added (new Fig. 6a).

5. Organoid experiments: The 3D assay in figure 6 is quite convincing. The authors may want to confirm results from earlier 2D culture experiments (Fig. 2-4) using this assay to strengthen their findings. One potential issue with the organoid data is that there appears to be a discrepancy between RepSox treated organoid size between Figure 6f and 6g. Are all SG organoid sizes within a group consistent? Please clarify. It may be worthwhile to quantify organoid size re: Figure 6b.

As Reviewer #2 acknowledges, variability in organoid sizes within a condition is a common pitfall of organoid research. We have now therefore added quantification of organoid size (new Fig. 7b and Supplementary Fig. 8a).

6. Page 12, pSMAD analysis: Why did the authors go back to 2D? What would it be like in 3D, which better recapitulates native SGs?

New pSMAD analysis in sebaceous organoids has been added (new Fig. 7h). These findings are consistent with the results obtained with 2D-cultured cells.

Readability Suggestions:

1. To help the non-expert reader, add a Figure 1a and b, prior to presenting your IHC images, with schematic images of: a) a pilosebaceous unit, annotating each compartment (e.g. cartoon with labeling of INF, IFE, JX, SG, HF/SH, and clear demarcation of the 'upper' pilosebaceous unit), and b) sebaceous gland development and structure (e.g. proliferative cells and lipid-rich sebocytes, upper or lower SG, sebaceous ducts, at early and late stages, ending with sebum release)

A new cartoon has been added (new Fig. 1a).

2. Again, in Figure 6 (or a last Figure), it would be helpful to see an overall summary schematic image of mechanism/signaling pathway that describes loss of GATA6 leading to acne (e.g. low vs high GATA6 -> TGFbeta -> pSMAD2/3 -> acne vs normal + immune system regulation; i.e. every signaling/mechanism involved in acne pathogenesis that is investigated in the manuscript)

A cartoon of the proposed mechanisms has now been included (new Fig. 8).

3. More white space in Figures 2-6 would be helpful.

The figures have been reformatted to include the new data, and improve legibility.

4. For all IHC data, the overlap of Cyan (nuclear protein) and White (DAPI) is very difficult to interpret. Consider a different color combination.

The colours of all the immunofluorescent images have been changed to red-green-blue.

5. Double check labels for consistency between text and figures: e.g. page 12 "CK7" and Fig 6g "KRT7"

These have been corrected.

6. Page 6, 2nd line: "Furthermore, GATA6 expression was drastically reduced..." In which condition/anatomical location?

This has been corrected to "was reduced in comedone walls".

7. Page 6, 6-9 lines: "...GATA6 expression was decreased [IN PATIENTS?], although the difference was only significant in the larger dataset".

This has been corrected to "was decreased in acne skin".

8. Double check all the figure numbers referenced in the text (e.g. page 6, last sentence: Supplementary Fig 1j. doesn't seem correct).

This has been corrected.

9. Supplementary Figure 2b and c: label concentration of Dox

This has been corrected in new Supplementary Fig. 3a-b.

10. Page 7, start of the section "GATA6 mediates the junctional zone and xxx": Add a first sentence describing the reason for testing the treatment of TRO w/wo LG.

This has been added.

11. Supplementary Figure 3: Add a high magnification image of TRO+LG next to the existing TRO+LG image to provide the reader with a clearer impression of a differentiated SebE6E7 cell with lipid vacuoles.

This has been added to new Supplementary Fig. 6d.

12. Fig. 3a and d: Change colors; IVL and LOR (also, KRT7 and PLET1) both are cytosolic staining that if any cells express both, it cannot be clarified and not convincing with current colors. Consider changing to red and green.

The colours of all immunofluorescent images have been changed to red-green-blue.

13. Fig 3d: The intensity of KRT7 and PLET1 both look dramatically higher in the GATA6 overexpressed condition. Any explanation or analysis of intensity comparison available?

GATA6 overexpression increases KRT7 and PLET1 expression in sebocytes. Quantification is provided (Fig. 4d).

14. Make sure to label all treatments in detail for clear understanding: e.g. Fig. 3f and Fig. 4e; label as "+Dox +Vehicle" & "+Dox +TRO+LG" and "+Dox +Vehicle" & "+Dox +RA 5uM".

These labels have been corrected in the new Fig. 4f, 5e-f, 7f-h and Supplementary Fig. 8a.

15. Page 11, end of first paragraph, "our results suggest a new...": This is a vague sentence, please inject more detail (e.g. ...upper pilosebaceous unit "via activation/suppression of IFN γ /HLA-DR/CD40").

We have now revised this sentence to detail the significant changes induced by GATA6 and discuss its role.

16. Page 12, 2nd paragraph, "Immunostaining of sebaceous organoids for PLET1 revealed similarities between...": The comparison is not clear in this sentence—exactly which groups are you comparing (shGATA6 expressing vs. shGATA6+RepSox? Or control+RepSox? Or all of RepSox treated groups?)—either way, they look different. shGATA6 size is smaller than all RepSox treated groups). In the following sentence, "controls" and "they" do not clearly indicate which groups is being referenced (comparison of each RepSox treated groups to each vehicle treated groups? Or all RepSox treated groups to vehicle treated control group?). Please clarify.

We apologize for this lack of clarity and have changed the paragraph accordingly. We have also added quantification of organoid size (new Fig. 7b and Supplementary Fig. 8a).

17. Fig 6f & g: Re-label the treatments (e.g. +Dox +Vehicle, +Dox +RepSox).
These labels have been corrected in the new Fig. 7f-g.

Reviewer #3:

This is a very good piece of work, which suggests that GATA6 is a key regulator of homeostasis in the upper part of the pilosebaceous unit using several in tissue and in vitro techniques including functional studies. The authors provide strong evidence for their conclusions. The most part of the work is novel and of extreme importance to scientists in the specific field. Moreover, it is interesting to researchers in other related disciplines.

On the other side, the authors claim that GATA6 may be involved in the development of acne lesions and an actionable target in the treatment of acne. Despite some well worked confirming data, Swanson JB et al have already shown that loss of Gata6 causes dilation of the hair follicle canal and sebaceous duct (Swanson JB et al. Loss of Gata6 causes dilation of the hair follicle canal and sebaceous duct. *Exp Dermatol* 28:345-349, 2019). Such findings can be seen in seborrheic skin but not obligatorily in acne. Similar findings are also described in exogenously induced acne (e.g. dioxin-induced acne) and in syndromes of aged, sun exposed skin (e.g. Favre-Racouchot disease).

In contrast, acne is also associated with reduction of the free hair follicle canal due to hyperkeratosis and not only with a secondarily occurring dilation of the hair follicle canal.

We thank Reviewer #3 for his/her comments and agree that acne comedones are not only dilated hair follicles, but also follicles with increased keratinisation and proliferation.

Although we agree that there is no valid mouse model for acne, we mentioned the study by Swanson et al. as we thought that it was indicative of an eventual phenotype in humans. We have now added new sections from acne patients in revised Fig. 2a and Supplementary Fig. 2; these show acne lesions at different stages have a reduced expression of GATA6.

Moreover, GATA6 has been shown in previous work of the same authors and others (Oulès B et al. Mutant Lef1 controls Gata6 in sebaceous gland development and cancer. *EMBO J* 38(9), pii: e100526, 2019 / Siltanen S et al. Transcription factor GATA-6 is expressed in malignant endoderm of pediatric yolk sac tumors and in teratomas. *Pediatr Res* 54(4):542-546, 2003) to be involved in tumorigenesis. How acne and skin tumour induction by the same gene can be associated, when no tumour comorbidity exists in acne?

We thank Reviewer #3 for asking this question. We previously identified GATA6 as a marker of sebaceous differentiation in sebaceous, benign and malignant, mouse and human tumours. Siltanen et al found that GATA6 was expressed in the differentiated sebaceous compartment of teratomas, reinforcing our own conclusions.

In addition, GATA6 knockdown in DMBA-treated K14 Δ NLef1 mice led to an increased rate of tumour formation and increased number of tumours per mouse. This led us to postulate that GATA6 is a tumour suppressor during sebaceous carcinogenesis, likely by controlling DNA mismatch repair response mechanisms.

As also suggested by the missense mutations in the GATA6 gene observed in human sebaceous tumours, we believe chronic GATA6 loss may be associated with tumour progression along with other additional mutations. However, there is no evidence in the literature that GATA6 loss *per se* could be directly involved in skin tumour initiation, thus explaining the absence of tumour comorbidity in acne.

This point has now been added to the Discussion.

Page 3, line 19ff: The mentioning of four pathogenetic factors of acne represents an old not any more valid theory of acne pathogenesis. The authors should refer to the current knowledge of acne pathogenesis (Zouboulis CC, Dessinioti C. A new concept of acne pathogenesis. In: Zouboulis CC et al (eds) Pathogenesis and Treatment of Acne and Rosacea. Springer, Berlin Heidelberg New York, pp 105-107, 2014).

The Introduction has been revised to provide a better overview of the current acne field and proposed pathobiology models.

Page 3, line 23; page 9, line 24; page 10, lines 12, 15, 16; page 14, line 17: "...Cutibacterium acnes (C. acnes) species (previously Propionibacterium acnes)." There is no obligatory, clinically relevant change of the terminology of Propionibacteria and, therefore, there is no need of substitution of the term "Propionibacterium acnes" by any other term (Alexeyev OA et al. Why we continue to use the name Propionibacterium acnes. Br J Dermatol 179:1227, 2018).

We have reverted back to *Propionibacterium acnes* and stated that this bacterium is also called *Cutibacterium acnes* by some authors as a matter of clarity for our readers.

Insights into the development and function of the pilosebaceous unit." The development of the pilosebaceous unit is also described in humans long ago. Morphogenesis and human sebaceous differentiation have been repeatedly reported (e.g. Zouboulis CC et al. Sebaceous Glands. In: Hoath SB, Maibach HI (eds) Neonatal Skin – Structure and Function. 2nd ed, Marcel Dekker, New York Basel, pp 59-88, 2003; Liakou AI et al. Marked reduction of number and individual volume of sebaceous glands in psoriatic lesions. Dermatology 232:415-424, 2016). The mouse-originating information on molecular pathways may provide additional functional information at the molecular level, in case that the mouse data would be shown in the future to be representative for human functions.

We apologize for this error. We now state in the Introduction that early work described the morphogenesis of pilosebaceous units during human embryology, citing Zouboulis CC et al Sebaceous Glands. In: Hoath SB, Maibach HI (eds) Neonatal Skin – Structure and Function. 2nd ed, Marcel Dekker, New York Basel, pp 59-88, 2003.

Page 4, line 12: Not retinoids but specifically isotretinoin is the most effective treatment for acne vulgaris and its effectiveness is not currently detected but since 40 years.

We thank the reviewer for this correction. We have specified that isotretinoin is the most effective treatment for acne vulgaris.

Page 4. Line 13: The ref. #22 has to be substituted by a more specific one, e.g. Zouboulis CC. Isotretinoin revisited – Pluripotent effects on human sebaceous gland cells. J Invest Dermatol 126:2154-2156, 2006 / Zouboulis CC. The truth behind this undeniable efficacy – Recurrence rates and relapse risk factors of acne treatment with oral isotretinoin. Dermatology 212:99-100, 2006 / Ganceviciene R, Zouboulis CC. Isotretinoin: state of the art treatment for acne vulgaris. Expert Rev Dermatol 2:693-706, 2007.

We have substituted the previous reference and now cite Zouboulis CC: 'The truth behind this undeniable efficacy – Recurrence rates and relapse risk factors of acne treatment with oral isotretinoin'. Dermatology 212:99-100, 2006.

Page 4, lines 17-18: "Since GATA6 plays a role in homeostasis of the upper HF in mouse, in the present report we have explored the potential link between GATA6 and acne pathogenesis." The consequence cannot be considered as straight ahead, since the mouse does

not develop acne even if GATA6 may play a role in homeostasis of its upper HF. None of the clinical characteristics of mouse HF changes resembles follicular inflammation in humans. Therefore, the authors should present another argumentation for their decision to study GATA6 expression changes and acne.

We have now clarified this section and state clearly that there is no valid mouse model to study acne. However, the role of GATA6 in the homeostasis of the upper HF in mice prompted us to explore a possible physiological role for GATA6 in the human upper hair follicle. Moreover, we postulated that GATA6 could be involved in the differentiation process of INF/JZ/SD and SG, and could play a central role in the “comedone switch” hypothesis of Jean-Hilaire Saurat (Saurat Dermatology 2015). This is why we sought to investigate the potential link between GATA6 and acne pathogenesis.

Page 5, line 23: Please use the term “Ki67” instead of “KI67”.

This has been corrected.

Page 6, line 3: Were the comedone-bearing patients not patients with acne vulgaris? Acne comedonica is a type of acne vulgaris, however, several disorders with comedones exist, which are acneiform diseases but not acne vulgaris. What was the correct diagnoses of the 4 patients?

We thank Reviewer #3 for this point. We removed the previous acne samples as there was a doubt about the correct diagnosis for one of them and because we wanted to process and stain all the patients' 3 sections at the same time to reduce technical variability (cf. Reviewer#2 specific comment 1). We therefore collected new sections and analysed 8 healthy and 10 acne vulgaris patients presenting with different disease severities (New Fig. 2a and Supplementary Fig. 2).

Page 7, section “GATA6 mediates the effects of retinoic acid on sebocytes”: It is a pity that the authors only have used all-trans-RA in their experiments. There is, indeed evidence that 13-cis-RA (isotretinoin) has to be intracellularly metabolized to all-trans-RA to become active on human sebocytes (Tsukada M et al. 13-cis Retinoic acid exerts its specific activity on human sebocytes through selective intracellular isomerization to all-trans retinoic acid and binding to retinoid acid receptors. J Invest Dermatol 115:321-327, 2000) but since isotretinoin is the most potent systemic drug for acne, the detection of isotretinoin effects similar to those detected by all-trans-RA would better corroborate the conclusion of the authors.

We used all-trans-RA because it is the active metabolite of isotretinoin and we did not know whether the isomerisation process would be as efficient in our cell model as in SZ95 cells (Tsukada et al J Invest Dermatol 2000).

However, we have now clarified in the text that isotretinoin (13-cis-retinoic acid) undergoes intracellular isomerisation to all-trans retinoic acid which then binds to RAR (Tsukada et al J Invest Dermatol 2000) so that the readers will understand the rationale for directly using all-trans-RA.

Page 9, section “Immune regulation of the upper pilosebaceous unit is controlled by GATA6”: In all previous articles using human sebocytes as targets or partners of inflammatory response arachidonic acid has been used as trigger factor and not PGN. Why the authors made the PGN experiments, which cannot be compared with any previous studies?

We decided to use peptidoglycan as it is a component of the *P.acnes* membrane, and therefore a good way to mimic the *P.acnes* effect on sebocytes. In addition, we found several studies in

which PGN was applied to SZ95 sebocytes to study the inflammatory response (e.g. Nguyen et al Scientific Rep 2018, Ke et al Chin J Dermatol 2019, Hou et al Innate Immunity 2019).

Page 9, line 24; page 14, line 17: “To mimic *C. acnes* infection, ...”: Acne is not an infectious disease and *P. acnes* does not induce an infection in acne (Xu DT et al. Is human sebum the source of skin follicular ultraviolet-induced red fluorescence? A cellular to histological study. *Dermatology* 234:43-50, 2018).

We apologize for this error and have changed the sentence accordingly.

Page 11, lines 8-9; page 12, lines 23-24: “For this reason, we developed a 3D sebaceous organoid model to better recapitulate SG homeostasis” and “By developing a new sebaceous organoid model, ...”: The described organoids have been previously published in Yoshida GJ et al. Three-dimensional culture of sebaceous gland cells revealing the role of prostaglandin E2-induced activation of canonical Wnt signaling. *Biochem Biophys Res Commun* 438:640–646, 2013.

We have now clarified this point by specifying that we developed a 3D sebaceous organoid model using SebE6E7 sebocytes (SZ95 sebocytes were used in Yoshida et al). We also changed the sentence “By developing a new sebaceous organoid model” to “By developing a sebaceous organoid model”.

Page 13, lines 4-6: “...but limits, in parallel, cell proliferation and lipid production to prevent hyperseborrhoea. It also orients sebocytes towards an anti-inflammatory phenotype.”: If GATA6 can, indeed, reduce lipogenesis and induce an anti-inflammatory phenotype in acne, it should also modify the lipid fractions produced by the human sebocytes from pro-inflammatory to anti-inflammatory ones (Zouboulis CC et al. Acne is an inflammatory disease and alterations of sebum composition initiate acne lesions. *J Eur Acad Dermatol Venereol* 28:527-532, 2014 / Töröcsik D et al. Leptin promotes a pro-inflammatory lipid profile and induces inflammatory pathways in human SZ95 sebocytes. *Br J Dermatol* 171:1326-1335, 2014 / Lovászi M et al. Sebaceous immunobiology is orchestrated by sebum lipids. *Dermatoendocrinol* 9:e1375636, 2017). Such experiments should be additionally performed to corroborate this conclusion.

This is indeed a very interesting point that we wish to further explore in the future by doing a dedicated study. We believe that it is beyond the scope of the current study.

Page 13, line 9: “Acne mostly occurs in adolescents” reads more correctly than “Acne is a disease that occurs in teenagers when puberty is initiated.”

This has been corrected.

Page 13, lines 11-12: “...the excellent therapeutic effect of combined oral contraceptive pills.” Acne only responds to hormonal antiandrogens and not to every contraceptive pill (Dessinioti C, Zouboulis CC. Hormonal therapy for acne. In: Zouboulis CC et al (eds) *Pathogenesis and Treatment of Acne and Rosacea*. Springer, Berlin Heidelberg New York, pp 477-482, 2014).

This has been corrected and we have changed the reference to the one you suggested.

Page 13, lines 12-14: “...patients with acne frequently have clinical signs of hyperandrogenism and increased serum and skin levels of testosterone and associated androgens such as dehydroepiandrosterone.” This is not true. Patients with acne have in maximum 20% clinical signs of hyperandrogenism but rarely hyperandrogenemia (Chen W,

Zouboulis CC. Acne and androgens. In: Zouboulis CC et al (eds) Pathogenesis and Treatment of Acne and Rosacea. Springer, Berlin Heidelberg New York, pp 131-134, 2014).

We thank Reviewer #3 for this correction. The sentence has been changed accordingly.

Page 14, lines 23-24: “This suggests that immune overactivation is detrimental in acne and that anti-inflammatory treatments are needed.”: Indeed, anti-inflammatory but not antibacterial (Zouboulis CC. Is acne vulgaris a genuine inflammatory disease? Dermatology 203:277-279, 2001 / Zouboulis CC. Zileuton, a new efficient and safe systemic anti-acne drug. Dermatoendocrinol 1:188-192, 2009).

We agree with Reviewer #3 and have added the reference Zouboulis CC: ‘Is acne vulgaris a genuine inflammatory disease?’ Dermatology 203:277-279, 2001.

Page 15, line 3-5: “In conclusion, our work identifies GATA6 as a crucial regulator of the upper pilosebaceous unit in human skin. Its loss is likely to contribute to the pathogenic features of acne, opening up new avenues for acne research and treatment.” Indeed, this work clearly shows a major role of GATA6 in the proper anatomy and function of the acroinfundibulum. It also indicates that GATA6 may be involved in acne. However, previous work of the same authors and others (Oulès B et al. Mutant Lef1 controls Gata6 in sebaceous gland development and cancer. EMBO J 38(9), pii: e100526, 2019 / Siltanen S et al. Transcription factor GATA-6 is expressed in malignant endoderm of pediatric yolk sac tumors and in teratomas. Pediatr Res 54(4):542-546, 2003) indicate an association of GATA6 downregulation with tumorigenesis. How acne and skin tumour induction by the same gene can be associated, when no tumour comorbidity exists in acne?

We have now clarified this very important point above, as well as in the Discussion.

Reference #4: The correct designation of the reference is: Moradi-Tuchayi, S. M. et al. Acne vulgaris. Nat. Rev. Dis. Prim. 1, 15029 (2015).

We apologize for this error. We have now corrected the reference.

REVIEWERS' COMMENTS:

Reviewer #1 (Remarks to the Author):

I commend the authors for improving their study substantially and for providing additional support for their key claims. The remaining limitations of their work, which the authors acknowledge in their rebuttal, but only incompletely in the revised manuscript, strike me as acceptable - if these are openly disclosed in the re-revised manuscript (I recommend appropriate text editing to do so more fully).

With these limitations in mind, I strongly suggest that the authors mitigate these two key claims (also in the Title, Abstract and Discussion component of their paper), as even the additional data provided do not yet sufficiently support these overreaching claims:

- a. "Loss of GATA6 contributes to acne pathogenesis in human skin"
- b. "the current work identifies GATA6 as an essential regulator of the human upper pilosebaceous unit in humans"

 The current evidence for these claims remains circumstantial and is not as definitive (i.e., supported by functional studies in models that can be accepted as sufficiently representative of human acne vulgaris or the distal pilosebaceous unit in human skin, respectively) as claimed here - not the least since the suggested human acne skin organ culture and xenotransplant experiments could not be performed, while the important HF organ culture gene silencing results remained somewhat inconclusive. Therefore, the authors should limit their claims to what one can reasonably conclude from phenomenological evidence in a limited number of human acne skin specimen, and functional evidence in models and cell lines whose relevance for human acne pathogenesis remains, at best, arguable.

Mitigating these claims does not significantly devalue the importance of this study, as everyone in the acne and pilosebaceous unit research field should be aware of the methodological limitations one faces and can be expected to understand the potential relevance of the study outcomes for human acne. Instead, overstating what can reasonably be concluded from the actual data provided and the limited models employed only detracts from the study's attractiveness.

Ralf Paus

Reviewer #2 (Remarks to the Author):

The authors have address all of my major concerns. I think the new version of the manuscript is much improved and in much better shape for publication. This will be a solid contribution to the field.

Reviewer #3 (Remarks to the Author):

The authors have answered the queries of the reviewers by performing additional experiments and modifying the text of the manuscript in a satisfactory manner.

Two references have to be corrected:

5. Schneider, M. R. & Schmidt-Ullrich, R. The Hair Follicle as a Dynamic Miniorgan. *Curr. Biol.* 19, R132–R142 (2009).

10. Zouboulis, C. C., Fimmel, S., Ortmann, J., Turnbull, J. R. & Boschnakow, A. Sebaceous Glands. in *Neonatal Skin – Structure and Function*. 2nd ed. 59–88 (Marcel Dekker, New York Basel 2003). doi: 10.1201/9780203911716.ch14

REVIEWERS' COMMENTS:

Reviewer #1 (Remarks to the Author):

I commend the authors for improving their study substantially and for providing additional support for their key claims. The remaining limitations of their work, which the authors acknowledge in their rebuttal, but only incompletely in the revised manuscript, strike me as acceptable - if these are openly disclosed in the re-revised manuscript (I recommend appropriate text editing to do so more fully).

With these limitations in mind, I strongly suggest that the authors mitigate these two key claims (also in the Title, Abstract and Discussion component of their paper), as even the additional data provided do not yet sufficiently support these overreaching claims:

- "Loss of GATA6 contributes to acne pathogenesis in human skin"
- "the current work identifies GATA6 as an essential regulator of the human upper pilosebaceous unit in humans"

 The current evidence for these claims remains circumstantial and is not as definitive (i.e., supported by functional studies in models that can be accepted as sufficiently representative of human acne vulgaris or the distal pilosebaceous unit in human skin, respectively) as claimed here - not the least since the suggested human acne skin organ culture and xenotransplant experiments could not be performed, while the important HF organ culture gene silencing results remained somewhat inconclusive. Therefore, the authors should limit their claims to what one can reasonably conclude from phenomenological evidence in a limited number of human acne skin specimen, and functional evidence in models and cell lines whose relevance for human acne pathogenesis remains, at best, arguable.

Mitigating these claims does not significantly devalue the importance of this study, as everyone in the acne and pilosebaceous unit research field should be aware of the methodological limitations one faces and can be expected to understand the potential relevance of the study outcomes for human acne. Instead, overstating what can reasonably be concluded from the actual data provided and the limited models employed only detracts from the study's attractiveness.

Ralf Paus

We thank Dr Paus for his comments and suggestions. We have revised the Abstract and Discussion to tone down our claims. We have, however, retained the original title because we feel we have provided robust evidence that GATA6 contributes to the homeostasis of the human upper pilosebaceous unit and that its loss plays a role in the pathological changes observed in acne.

Reviewer #2 (Remarks to the Author):

The authors have address all of my major concerns. I think the new version of the manuscript is much improved and in much better shape for publication. This will be a solid contribution to the field.

We thank Reviewer 2 for his/her comments and suggestions that have helped to improve our manuscript.

Reviewer #3 (Remarks to the Author):

The authors have answered the queries of the reviewers by performing additional experiments and modifying the text of the manuscript in a satisfactory manner.

Two references have to be corrected:

5. Schneider, M. R. & Schmidt-Ullrich, R. The Hair Follicle as a Dynamic Miniorgan. *Curr. Biol.* 19, R132–R142 (2009).

10. Zouboulis, C. C., Fimmel, S., Ortmann, J., Turnbull, J. R. & Boschnakow, A. Sebaceous Glands. in *Neonatal Skin – Structure and Function*. 2nd ed. 59–88 (Marcel Dekker, New York Basel 2003). doi: 10.1201/9780203911716.ch14

We thank Reviewer 3 for his comments and suggestions that have helped to improve our manuscript. We have corrected the two references.